# RNA-directed DNA methylation prevents rapid and heritable reversal of transposon silencing under heat stress in *Zea mays*

**Wei Guo**[1], **Dafang Wang**[2], **Damon Lisch**[1]*

**1** Department of Botany and Plant Pathology, Purdue University, West Lafayette, Indiana, United States of America, **2** Division of Math and Sciences, Delta State University, Cleveland, Mississippi, United States of America

* dlisch@purdue.edu

**Data Availability Statement:** All relevant data are within the manuscript and its Supporting Information files.

## Abstract

In large complex plant genomes, RNA-directed DNA methylation (RdDM) ensures that epigenetic silencing is maintained at the boundary between genes and flanking transposable elements. In maize, RdDM is dependent on *Mediator of Paramutation1 (Mop1)*, a gene encoding a putative RNA dependent RNA polymerase. Here we show that although RdDM is essential for the maintenance of DNA methylation of a silenced *MuDR* transposon in maize, a loss of that methylation does not result in a restoration of activity. Instead, heritable maintenance of silencing is maintained by histone modifications. At one terminal inverted repeat (TIR) of this element, heritable silencing is mediated via histone H3 lysine 9 dimethylation (H3K9me2), and histone H3 lysine 27 dimethylation (H3K27me2), even in the absence of DNA methylation. At the second TIR, heritable silencing is mediated by histone H3 lysine 27 trimethylation (H3K27me3), a mark normally associated with somatically inherited gene silencing. We find that a brief exposure of high temperature in a *mop1* mutant rapidly reverses both of these modifications in conjunction with a loss of transcriptional silencing. These reversals are heritable, even in *mop1* wild-type progeny in which methylation is restored at both TIRs. These observations suggest that DNA methylation is neither necessary to maintain silencing, nor is it sufficient to initiate silencing once has been reversed. However, given that heritable reactivation only occurs in a *mop1* mutant background, these observations suggest that DNA methylation is required to buffer the effects of environmental stress on transposable elements.

## Author summary

Most plant genomes are mostly transposable elements (TEs), most of which are held in check by modifications of both DNA and histones. The bulk of silenced TEs are associated with methylated DNA and histone H3 lysine 9 dimethylation (H3K9me2). In contrast, epigenetically silenced genes are often associated with histone lysine 27 trimethylation (H3K27me3). Although stress can affect each of these modifications, plants are generally competent to rapidly reset them following that stress. Here we demonstrate that although

**Funding:** This work was funded by a grant from the National Science Foundation to DL (IOS-1237931). https://www.nsf.gov/index.jsp. The funders had no role in study design, data collection and analysis, decision to publish, or preparation of the manuscript.

**Competing interests:** The authors have declared that no competing interests exist.

DNA methylation is not required to maintain silencing of the *MuDR* element, it is essential for preventing heat-induced, stable and heritable changes in both H3K9me2 and H3K27me3 at this element, and for concomitant changes in transcriptional activity. These finding suggest that RdDM acts to buffer the effects of heat on silenced transposable elements, and that a loss of DNA methylation under conditions of stress can have profound and long-lasting effects on epigenetic silencing in maize.

## Introduction

Transposable elements (TEs) are a ubiquitous feature of all genomes. They survive in large measure because they can out-replicate the rest of the genome [1]. As a consequence of that replication TEs can threaten the integrity of the host genome. In response to this threat, all forms of life have evolved mechanisms by which TEs can be silenced when they are recognized as such and, importantly, maintained in a silenced state over long periods of time, even when the initial trigger for silencing is no longer present [2–4]. Because plant genomes are largely composed of TEs, the majority of plant DNA is maintained in an epigenetically silent state [5]. Because they are the primary target of epigenetic silencing in plants, TEs are an excellent model for understanding the means by which particular DNA sequences are targeted for silencing, and for understanding the means by which silencing can be maintained from one generation to the next [6]. Finally, because TEs have proved to be exquisitely sensitive to a variety of stresses [7–9], they can also teach us a great deal about the relationship between stress and epigenetically encoded memory of stress response.

In plants, heritable epigenetic silencing of TEs is almost invariably associated with DNA methylation [10–12]. The vast bulk of TEs in plant genomes are methylated and, with some notable exceptions [13], epigenetically silenced [14,15]. DNA methylation has a number of features that makes it an appealing mechanism by which silencing can be heritably propagated, either following cell divisions during somatic development, or transgenerationally, from one generation to the next. Because methylation in both the CG and CHG sequence contexts (where H = A, T or G) are symmetrical, information concerning prior DNA methylation can be easily propagated by methylating newly synthesized DNA strands using the parent strand as a template. For CG methylation, this is achieved by reading the methylated cytosine using VARIANT IN METHYLATION 1–3 (VIM1-3) [16,17] and writing new DNA methylation using the methyl transferase MET1 [18–20]. For CHG, methylation is read indirectly by recognition of H3K9 dimethylation (H3K9me2) by CMT3, which catalyzes methylation of newly synthesized DNA, which in turn triggers methylation of H3K9 [21–23].

Maintenance methylation of most CHH involves RNA-directed DNA methylation (RdDM). The primary signal for *de novo* methylation of newly synthesized DNA from previously methylated DNA sequences is thought to be transcription by RNA POLYMERASE IV (POLIV) of short transcripts from previously methylated templates [24–26]. This results in the production of small RNAs that are tethered to the target DNA by RNA POLYMERASE V (POLV), which is targeted by SU(VAR)3-9 homologs SUVH2 and SUVH9, which bind to methylated DNA [27]. This in turn triggers *de novo* methylation of newly synthesized DNA strands using the methyl transferases DRMT1/2 [28,29]. In addition to the RdDM pathway, CHH methylation can also be maintained due to the activity of CHROMOMETHYLASE2 (CMT2), which, similar to CMT3, works in conjunction with H3K9me2 to methylate non-CG cytosines, particularly in deeply heterochromatic regions of the genome [30]. Finally, because both histones and DNA must be accessible in order to be modified, chromatin remodelers

such as DDM1 are also often required for successful maintenance of TE silencing [23,31]. In plants, effective silencing of TEs requires coordination between DNA methylation and histone modifications [32]. Together, these pathways can in large part explain heritable propagation of both DNA methylation and histone modification of TEs.

In large genomes such as that of maize, much of RdDM activity is focused not on deeply silenced heterochromatin, which is often concentrated in pericentromeric regions, but on regions immediately adjacent to genes, referred to as "CHH islands" because genes in maize are often immediately adjacent to silenced TEs [15,33]. In maize, mutations in components of the RdDM pathway affect both paramutation and transposon silencing [34]. Mutations in *Mediator of Paramutation 1 (Mop1)*, a homolog of *RNA DEPENDENT RNA POLYMERASE2 (RDR2)*, result in the loss of nearly all 24 nucleotide small RNAs, as well as the CHH methylation that is associated with them [35–37]. Despite this, *mop1* has only minimal effects on gene expression in any tissue except the meristem [33,38], and the plants are largely phenotypically normal [39]. This, along with similar observations in Arabidopsis, has led to the suggestion that the primary role of RdDM is to reinforce boundaries between genes and adjacent TEs, rather than to regulate gene expression [33]. However, it should be noted that the *mop1* mutation can in some cases have effects on plant phenotype [40]. Further, *mop1* mutants can enhance the effects of exogenously applied ABA [41] and mutants of *Required to maintain repression6* (*Rmr6*), a homolog of the PolIV subunit DNA-directed RNA polymerase IV subunit 1 (NRPD1) [42], are altered in their response to drought, suggesting that the RdDM pathway may play a role in buffering stress responses in maize [43,44]. Further, even in wild-type backgrounds, there is evidence that the process of heritable paramutation of an allele of *R1*, which is known to be dependent on RdDM, is sensitive to changes in temperature and light during specific stages of development [45].

Unlike animals, plants do not experience a global wave of DNA demethylation either in the germinal cells of the gametophyte or in the early embryo [46]. Thus, DNA methylation and associated histone modifications are an attractive mechanism for transgenerationally propagated silencing. Indeed, there is strong evidence that mutants that trigger a global loss of methylation can cause heritable reactivation of previously silenced TEs, although it is worth noting that even in mutants in which the vast majority of DNA methylation has been lost, only a subset of TEs are transcriptionally reactivated [47,48], and DNA methylation of many TEs can be rapidly reestablished at many loci via RdDM in wild-type progenies of mutant plants, suggesting that memory propagated via DNA methylation can be restored due to the presence of small RNAs that can trigger *de novo* methylation of previously methylated sequences [49,50].

In contrast to TEs, most genes that are silenced during somatic development in plants are associated with H3K27 trimethylation (H3K27me3), which requires the activity of the polycomb complexes PRC2 and PRC1, which together catalyze H3K27 methylation and facilitate its heritable propagation [51–53]. In plants, H3K27me3 enrichment is generally associated with genes rather than TEs [54,55], and numerous developmental pathways require the proper deposition and maintenance of this modification [56,57]. The most well explored example of this involves epigenetic setting of *FLOWERING LOCUS C* (*FLC*), a negative regulator of flowering in Arabidopsis [58,59]. In a process known as vernalization, prolonged exposure to cold results in somatically heritable silencing of this gene, which in turn results in flowering under favorable conditions in the spring. Somatically heritable silencing of *FLC* is initially triggered by non-coding RNAs, which are involved in recruitment of components of PRC2, which catalyze H3K27me3, which in turn mediates a somatically heritable silent state [58]. Importantly, H3K27me3 at genes like *FLC* is erased each generation, both in pollen and in the early embryo [60–62]. The fact that H3K27me3 must be actively reset suggests that in the absence of this

resetting, H3K27me3 in plants is competent to mediate transgenerational silencing but is normally prevented from doing so.

Dramatic differences in TE content between even closely related plant species suggest that despite the relative stability of TE silencing under laboratory conditions, TEs frequently escape silencing and proliferate in natural settings [63]. Stress, both biotic and abiotic can often trigger TE transcription and, at least in some cases, transposition [7,64–67]. Further, there is evidence that the association of TEs and genes can result in *de novo* stress induction of adjacent genes [64,68,69].

Because of its dramatic and global effects on both gene expression and protein stability, heat stress has attracted considerable attention, particularly with respect to heritable transmission of TE activity. Although heat stress can trigger somatically heritable changes in gene expression, there appear to be a variety of mechanisms to prevent or gradually ameliorate transgenerational transmission of those changes [70,71]. Thus, for instance, although the *ONSEN* retrotransposon is sensitive to heat, it is only in mutants in the RdDM pathway that transposed elements are transmitted to the next generation [9,72]. Given that various components of regulatory pathways that have evolved to regulate TEs are up-regulated in germinal lineages, it is not surprising that a defect in one of these pathways would lead to an enhancement in the number of germinally transmitted new insertions [73,74]. The observation that it is the combination of both heat and components of the RdDM pathway results in reactivation of TEs, rather than each by itself has led to the suggestion that a key role of RdDM is to prevent TE activation specifically under conditions of stress [9,75].

Similar experiments using silenced transgenes have demonstrated that double mutants of *mom1* and *ddm1* cause silenced transgenes as well as several TEs to be highly responsive to heat stress, and the observed reversal of silencing can be passed on to a subsequent generation, but only in mutant progeny [76]. It is also worth noting that in many cases of TE reactivation, silencing is rapidly re-established in wild-type progeny [77,78]. The degree to which this is the case likely depends on a variety of factors, from the copy number of a given element, its position within the genome, its mode of transposition and the presence or absence of trans-acting small RNAs targeting that TE [79].

Our model for epigenetic silencing is the *Mutator* system of transposons in maize. The *Mutator* system is a family of related elements that share similar, 200 bp terminal inverted repeats (TIRs) but that contain distinct internal sequences. Nonautonomous *Mu* elements can only transpose in the presence of the autonomous element, *MuDR*. *MuDR* is a member of the MULE superfamily of Class II cut and paste transposons [80,81]. In addition to being required for transposition, the 200 bp TIRs within *MuDR* elements serve as promoters for the two genes encoded by *MuDR*, *mudrA*, which encodes a transposase, and *mudrB*, which encodes a novel protein that is required for *Mu* element integration. Both genes are expressed at high levels in rapidly dividing cells, and expression of both of them is required for full activity of the *Mutator* system [82,83]. MURA, the protein produced by *mudrA*, is sufficient for somatic excision of *Mu* elements, which results in characteristically small revertant sectors in somatic tissue. *MuDR* elements can be heritably silenced when they are in the presence of *Mu killer (Muk)*, a rearranged variant of *MuDR* whose transcript forms a hairpin that is processed into 21–22 nt small RNAs that directly trigger transcriptional gene silencing (TGS) of *mudrA* and indirectly trigger silencing of *mudrB* when it is in trans to *mudrA* [4,84]. Because *Muk* can be used to heritably silence *MuDR* through a simple cross, and because silencing of *MuDR* can be stably maintained after *Muk* is segregated away, the *MuDR/Muk* system is an excellent model for understanding both initiation and maintenance of silencing. Prior to exposure to *Muk*, *MuDR* is fully active and is not prone to spontaneous silencing [85]. After exposure, *MuDR* silencing is exceptionally stable over multiple generations [84].

When *mudrA* is silenced, DNA methylation in all three sequence contexts accumulates within the 5' end of the TIR immediately adjacent to *mudrA* (TIRA) [86]. Methylation at the 5' and 3' portions of this TIR have distinctive causes and consequences. The 5' end of the TIR is readily methylated in the absence of the transposase, but this methylation does not induce transcriptional silencing of *mudrA* [87]. Methylation in this end of TIRA is readily eliminated in the presence of functional transposase. However, the loss of methylation in a silenced element in this part of the TIRA does not result in heritable reactivation of a silenced element. In contrast, CG and CHG methylation in the 3' portion of TIRA, which corresponds to the *mudrA* transcript as well as to *Muk*-derived 22 nt small RNAs that trigger silencing, is not eliminated in the presence of active transposase and is specifically associated with heritable transcriptional silencing of *mudrA*.

The second gene encoded by *MuDR* elements, *mudrB*, is also silenced by *Muk*, but the trajectory of silencing of this gene is entirely distinct, despite the fact that the *Muk* hairpin has near sequence identity to the TIR adjacent to *mudrB* (TIRB) [4,84]. By the immature ear stage of growth in $F_1$ plants that carry both *MuDR* and *Muk*, *mudrA* is transcriptionally silenced and densely methylated. In contrast, *mudrB* in intact elements remains transcriptionally active in this tissue, but its transcript is not polyadenylated. It is only in the next generation that steady state levels of transcript become undetectable. Further, experiments using deletion derivatives of *MuDR* that carry only *mudrB* are not silenced by *Muk* when they are on their own, or when they are in trans to an intact *MuDR* element that is being silenced by *Muk*. This suggests that heritable silencing of *mudrB* is triggered by the small RNAs that target *mudrA*, but the means by which this occurs is indirect and involves spreading of silencing information from *mudrA* to *mudrB*.

Silencing of *mudrA* can be destabilized by the *mop1* mutant. MOP1 is homolog of RDR2 that is required for the production of the vast bulk of 24 nt small RNAs in maize, including those targeting *Mu* TIRs [35–37,88]. However, silencing of *MuDR* by *Muk* is unimpeded in a *mop1* mutant background, likely because *Muk*-derived small RNAs are not dependent on *mop1* [89]. Further, although reversal of silencing of *MuDR* in a *mop1* mutant background does occur, it only occurs gradually, over multiple generations, and only affects *mudrA*. In contrast, *mudrB* is not reactivated in this mutant background and, because *mudrB* is required for insertional activity, although these reactivated elements can excise during somatic development, they cannot insert into new positions.

## Results

### DNA methylation is not required to maintain silencing of *MuDR* elements in *mop1* mutants

Given that *MuDR* elements are only activated after multiple generations in a *mop1* mutant background, we wanted to understand how silencing of *MuDR* is maintained in *mop1* mutants prior to reactivation. To do this, we examined expression and DNA methylation at TIRA by performing bisulfite sequencing of TIRA of individuals in families that were segregating for a single silenced *MuDR* element, designated *MuDR**, and that were homozygous or heterozygous for *mop1* (S1 Fig).

In control plants carrying an active *MuDR* element, all cytosines in TIRA were unmethylated, which was consistent with our previous results (Fig 1B). Also consistent with previous results, $F_2$ *MuDR**/-; *mop1*/+ plants, whose $F_1$ parent carried both *MuDR* and *Muk*, exhibited dense methylation at TIRA. In contrast, DNA methylation in the CG, CHH and CHG contexts at TIRA was absent in *mop1* mutant siblings. Interestingly, *mop1* had effects on TIRB that are more consistent with the known effects of this mutant specifically on CHH methylation. While $F_2$ *MuDR**/-; *mop1*/+ plants exhibited dense methylation at TIRB in all sequence contexts, *mop1* homozygous

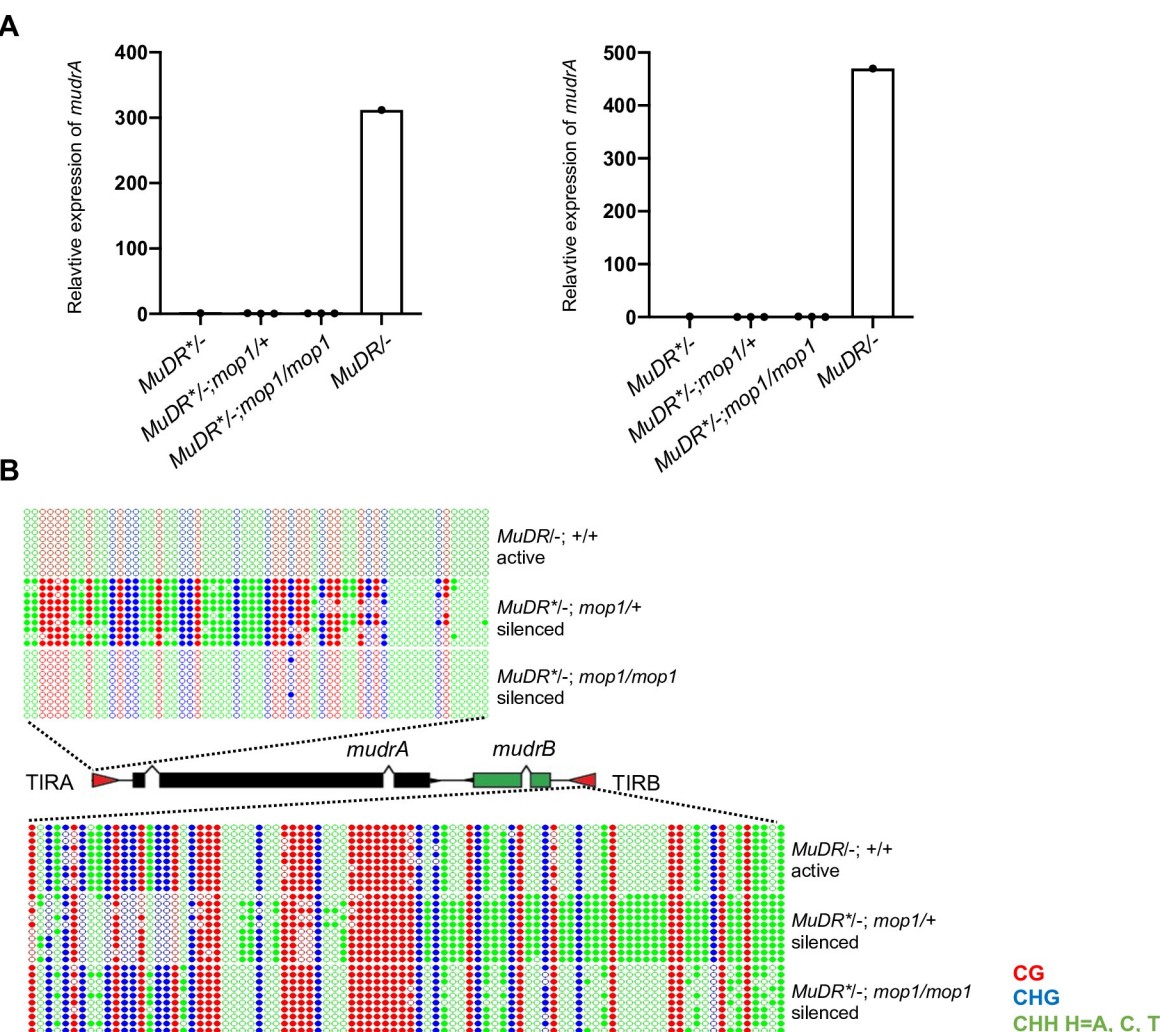

**Fig 1. DNA methylation patterns at TIRA and TIRB of stably silenced F₂ plants.** (A) qPCR analysis of *mudrA* and *mudrB* expression from *MuDR\*/-; mop1/+* and *MuDR\*/-;mop1/mop1* plants. *MuDR*: active element. *MuDR\**: inactive element. *Tub2* is used as an internal control gene. Six biological replicates are used for each experiment; two of six biological replicates are pooled together for each amplification. Error bars indicate mean ± standard deviation (SD) of three individuals. (B) DNA methylation patterns at TIRA and TIRB. Ten individual clones were sequenced from amplification of bisulfite-treated samples of the indicated genotypes. The cytosines in different sequence contexts are represented by different colors (red, CG; blue, CHG; green, CHH, where H = A, C, or T). For each genotype, DNA from six biological replicates were pooled.

siblings exhibited a loss of methylation only in the CHH context. Despite the effects of *mop1* on *MuDR* methylation at both TIRA and TIRB, qRT-PCR results demonstrated that these *mop1* mutant plants did not exhibit reactivation of *mudrA* or *mudrB* (Fig 1A).

## A loss of MOP1 enhances enrichment of H3K9 and H3K27 dimethylation at TIRA

Transposon silencing is often associated with H3K9 and H3K27 dimethylation, two hallmarks of transcriptional silencing in plants [21,55]. DNA methylation, particularly in the CHG context, is linked with H3K9me2 through a self-reinforcing loop, and these two epigenetic marks often colocalize at TEs and associated nearby genes [90]. We had previously demonstrated that these two repressive histone modifications corresponded well with DNA methylation of

silenced *MuDR* elements at TIRA [86]. However, our observation that silencing of *mudrA* can be maintained in the absence of DNA methylation in *mop1* mutants suggests that additional repressive histone modifications may be responsible for maintaining the silenced state of *mudrA*. To test this hypothesis, we examined the enrichment of H3K9me2 at TIRA in individuals in a family that segregated for silenced *MuDR* and for *mop1* homozygotes and heterozygotes by performing a chromatin immunoprecipitation quantitative PCR (ChIP-qPCR) assay. As controls, we also examined these two histone modifications in leaf tissue from plants carrying active and deeply silenced *MuDR* elements in a wild-type background. Compared with active *MuDR/-; +/+* plants, H3K9me2 and levels were significantly enriched at TIRA in the *MuDR*\*/-; +/+* plants (Fig 2A). The same was true of H3K27me2 (S2 Fig). Surprisingly, a

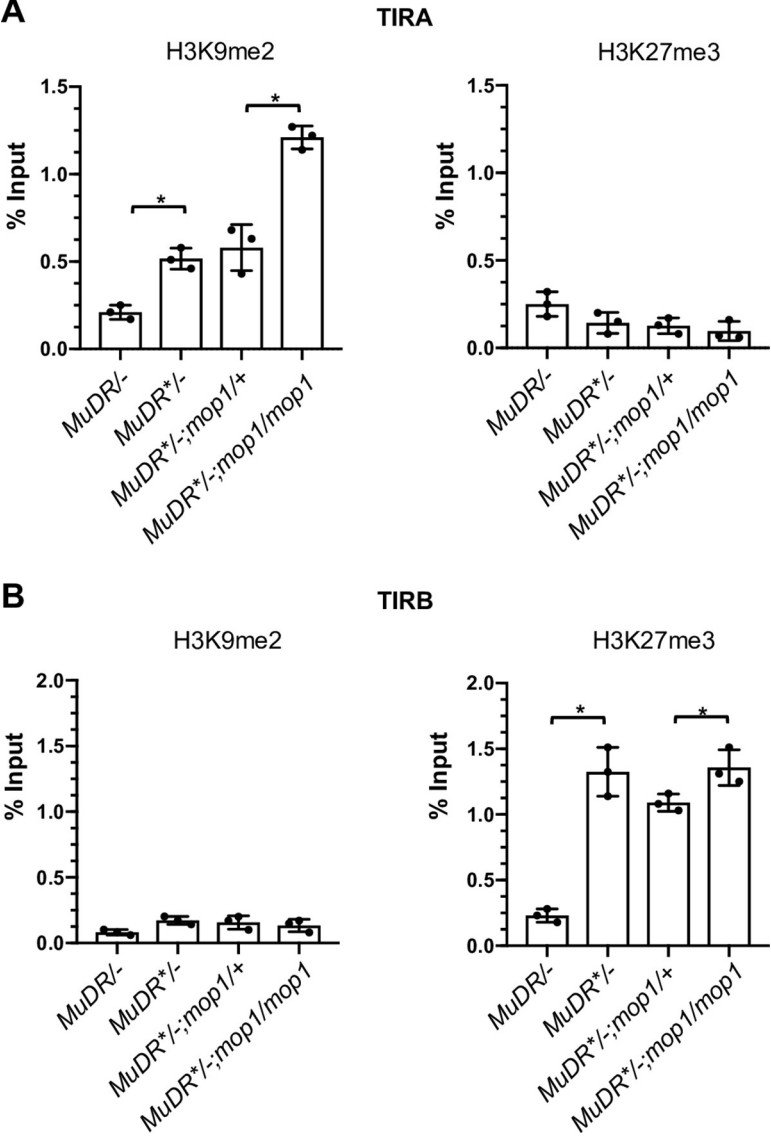

**Fig 2. ChIP-qPCR analysis of enrichment of histone marks H3K9me2 and H3K27me3 at TIRA and TIRB in *mop1* mutants.** ChIP-qPCR analysis of enrichment of histone marks, H3K9me2 and H3K27me3 at TIRA and TIRB. (A) Relative enrichment of H3K9me2 and H3K27me3 in leaf 3 of plants of the indicated genotypes. *MuDR*: active element. *MuDR*\*: inactive element. (B) Relative enrichment of H3K9me2 and H3K27me3 in leaf 3 of plants of the indicated genotypes. qPCR signal was normalized to *Copia* and then to the value of input sample. An unpaired t-test was performed. Error bars indicate mean ± standard deviation (SD) of the three biological replicates. *P<0.05; **P < 0.01

significant increase in H3K9me2 and H3K27me2 at TIRA was observed in *mop1* mutants compared with their *mop1* heterozygous siblings and with the silenced *MuDR*\*/-; +/+ control plants, suggesting that the loss of DNA methylation that resulted from the loss of MOP1 in these mutants actually resulted in an increase in both of these repressive chromatin marks.

## Silencing of TIRB is associated with an increase in H3K27me3

Like *mudrA*, *mudrB* is silenced by *Muk*, but maintenance of *mudrB* silencing has distinct requirements. Unlike *mudrA*, which is eventually reactivated in a *mop1* mutant background under normal conditions, *mudrB* remains silenced, suggesting that maintenance of silencing of this gene is independent of MOP1 [36]. ChIP-qPCR revealed that silencing of *mudrB* is not associated with H3K9me2 methylation. Instead, heritably silenced TIRB is enriched for H3K27me3, a modification normally associated with somatically silenced genes rather than transposable elements (Fig 2B). The *mop1* mutant appears to enhance H3K27me3 at TIRB relative to the *mop1* heterozygous siblings, although the enrichment is no greater that observed in the *MuDR*\*/-; +/+ controls.

## Application of heat stress specifically in the early stage of growth can promote the reactivation of silenced *MuDR* elements in *mop1* mutants

There is ample evidence that a variety of stresses can reactivate epigenetically silenced TEs. One particularly effective treatment is heat stress. Given that a loss of methylation by itself is not sufficient to reactivate silenced *MuDR* elements, we subjected *mop1* mutant and *mop1* heterozygous sibling seedlings carrying silenced *MuDR* elements (*MuDR*\*) to heat stress. Fourteen-day-old *MuDR*\*/-; *mop1/mop1* and *MuDR*\*/-; *mop1/+* sibling seedlings were heated at 42°C for four hours and leaf samples were collected immediately after that treatment (Fig 3A). qRT-PCR for the heat response factor *Hsp90* (Zm00001d024903) confirmed that the seedlings were responding to the heat treatment (S3 Fig). We then examined *MuDR* transcription by performing qRT-PCR on RNA from leaf three immediately after the plants had been removed from heat and from control plants that had not been subjected to heat stress. In the *mop1* mutants, both *mudrA* and *mudrB* became transcriptionally reactivated upon heat treatment (Fig 3B). *MuDR* elements in plants that were *mop1* mutant that were not heat stressed and were those that were wild-type and that were heat stressed were not reactivated, demonstrating that both a mutant background and heat stress are required for efficient reactivation. To determine if the application of heat stress at a later stage of plant development can also promote reactivation, we heat-stressed 28-day-old plants and examined *MuDR* transcription in leaf seven at a similar stage of development (~10 cm) as had been examined in heat stressed leaf three in the previous experiment. In these plants, we saw no evidence of *MuDR* reactivation although qRT-PCR *Hsp90* indicated that these seedlings were responding to the heat treatment (Figs 3B and S3). Taken together, these data suggest that the application of heat stress specifically at an early stage of plant development can promote the reactivation of a silenced TE in a mutant that is deficient in the RdDM pathway.

TIRA in a *mop1* mutant background already lacks any DNA methylation prior to heat treatment and thus heat would not be expected to reduce TIRA methylation. However, in *mop1* mutants TIRB retained CG and CHG methylation and also remained inactive (Fig 1B). To determine if reactivation after heat treatment is associated with a loss of this methylation, we examined DNA methylation at TIRB in *mop1* mutants in the presence or absence of heat treatment. This assay was performed on the same tissues that we collected for *MuDR* expression reactivation analysis. We found that the DNA methylation pattern was the same for both the heat-treated and the control *mop1* mutant plants, indicating that heat stress does not alter

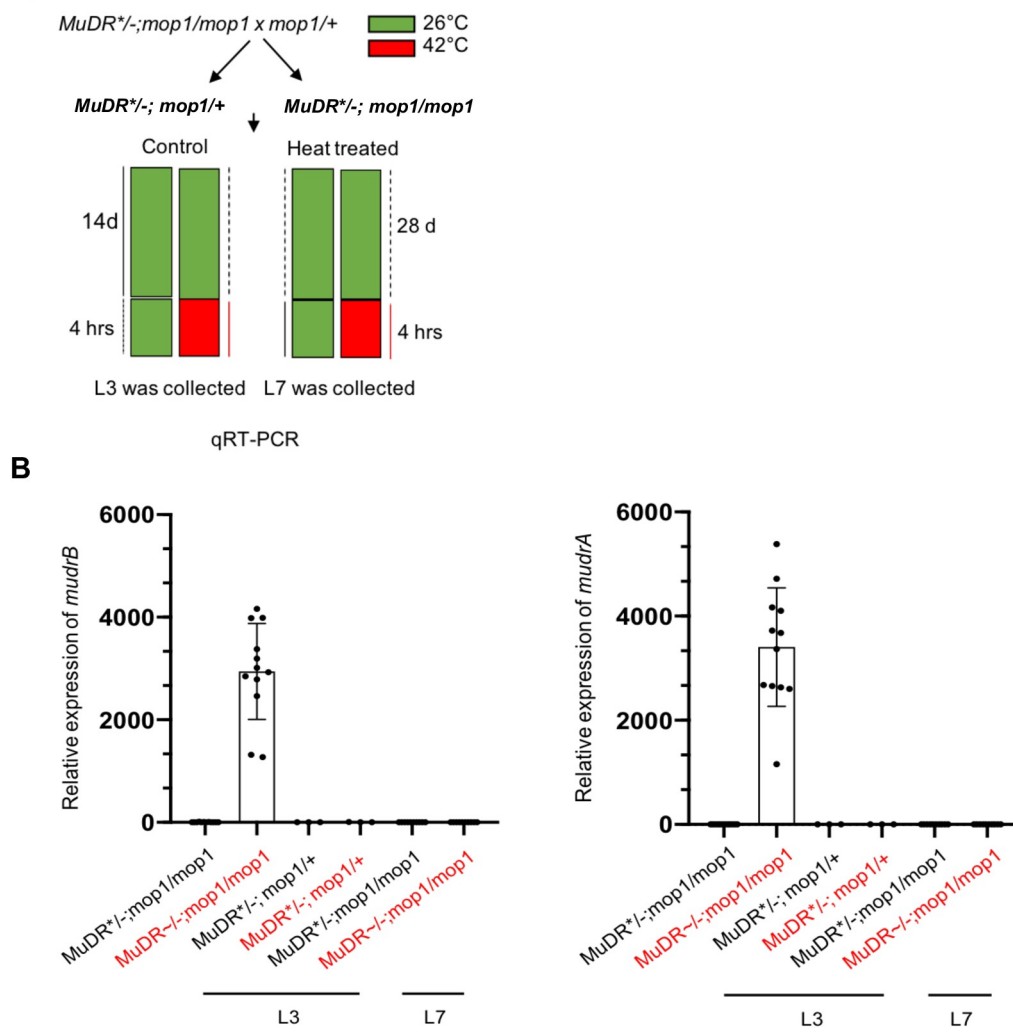

**Fig 3. Expression of *mudrA* and *mudrB* in plants under heat stress.** (A) Schematic diagram of the heat-reactivation experiment. (B) qRT-PCR of *mudrA* and *mudrB* in leaf 3 and leaf 7 in plants of the indicated genotypes. Twelve biological replicates are used for each experiment. *Tub2* is an internal control gene. Additional controls for each experiment include *MuDR/-*, pooled samples from twelve heated *MuDR\*/-; mop1/+* plants and twelve unheated plants. Red text is used to indicate samples that were subjected to heat stress.

TIRB methylation and that a further loss of DNA methylation is not the cause of *mudrB* reactivation in this tissue (S4 Fig).

## Heat stress reverses TE silencing by affecting histone modifications at TIRA and TIRB

Under normal conditions, we found that H3K9me2 at TIRA is associated with silencing, and H3K9me2 is actually enriched when TIRA methylation is lost in *mop1* mutants (Fig 2A). In contrast, we find that H3K27me3, rather than H3K9me2, is enriched at TIRB and is maintained at similar or slightly elevated levels in *mop1* mutant relative to *mop1* heterozygous siblings (Fig 2B). Given these observations, we hypothesized that heat stress may reverse H3K9me2 enrichment at TIRA and H3K27me3 enrichment at TIRB. To test this hypothesis,

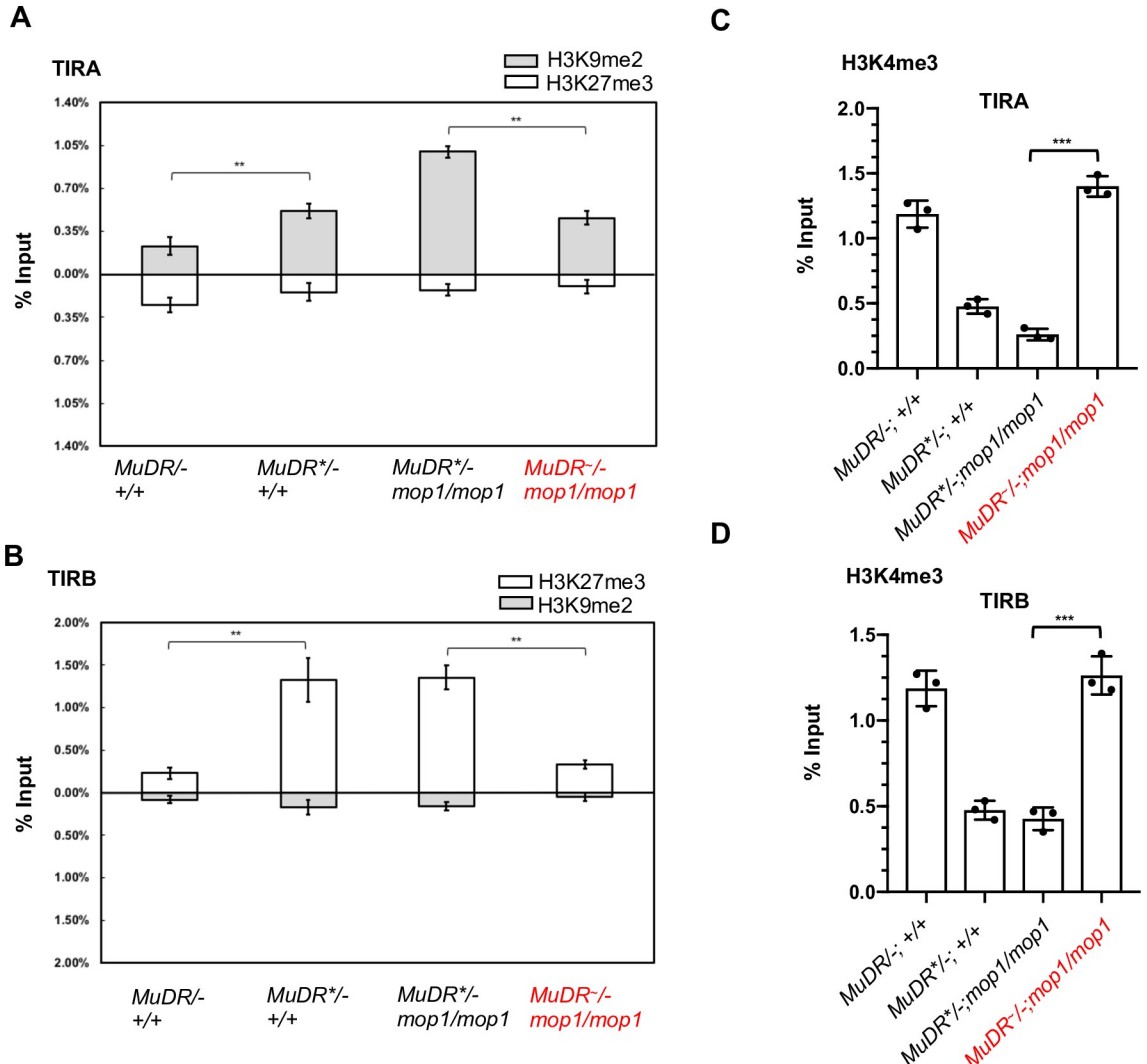

**Fig 4. ChIP-qPCR analysis of histone marks TIRA and TIRB under heat stress.** Relative enrichment of H3K9me2 and H3K27me3 at TIRA (A) and TIRB (B) in leaf 3 of plants of the indicated genotypes. (Relative enrichment of H3K4me3 at TIRA (C) and TIRB (D) in leaf 3 of plants of the indicated genotypes. qPCR signals were normalized to *Copia* and then to the value of input samples. *MuDR** refers to a silenced *MuDR* element. *MuDR~* refers to a reactivated element. Red text indicates a sample that has been heat-treated. Error bars indicate mean ± standard deviation (SD) of the three biological replicates. **P < 0.01; ***P < 0.001

we determined the level of H3K9me2 and H3K27me3 at TIRA and TIRB under normal and stressed conditions using ChIP-qPCR.

Upon heat stress, the level of H3K9me2 at TIRA was significantly decreased in *mop1* mutants compared to that of non-treated *mop1/mop1* mutant siblings (Fig 4A). Interestingly, however, H3K9me2 enrichment only decreased to the level observed at TIRA in silenced *MuDR**/-; +/+ plants, and it remained significantly higher than that of TIRA in the naturally active *MuDR/-; +/+* plants. In contrast, we observed no changes in H3K27me3 at TIRA.

At TIRB, we observed no changes in H3K9me2 enrichment in any of our samples. Instead, we found that heat treatment reversed previously established H3K27me3 at TIRB, supporting the hypothesis that this modification, rather than H3K9me2, mediates heritable silencing of *mudrB* (Fig 4B). Consistent with evidence for transcriptional activation of both *mudrA* and *mudrB*, we observed enrichment of the active mark histone H3 lysine 4 trimethylation

(H3K4me3) in reactivated TIRA and TIRB (Fig 4C and 4D). Taken together, these data demonstrate that heat stress can simultaneously reduce two often mutually exclusive repressive histone modifications, H3K9me2 and H3K27me3 at the two ends of a single TE.

### The reactivation state is somatically transmitted to the new emerging tissues

We next sought to determine whether or not the reactivated state can be propagated to cells in somatic tissues after the heat had been removed. We performed quantitative RT-PCR to detect *mudrA* and *mudrB* transcripts in mature leaf ten of plants 35 days after the heat stress and in immature tassels ten days after that. At V2, when the heat stress was applied and leaf three was assayed, cells within leaf 10 primordia are present and may have experienced the heat stress. In contrast, because the tassel primordia are not formed until V5, the cells of the tassel could not have experienced the heat stress directly [91,92]. We found that both genes stayed active in both tissues, indicating heat-induced reactivation is stably transmitted to new emerging cells and tissues (Fig 5).

### *MuDR* activity is stably transmitted to subsequent generations

Our previous work had demonstrated that silenced *mudrA* (but not *mudrB*) can be progressively and heritably reactivated only after multiple generations of exposure to the *mop1* mutation under normal conditions. Only after eight generations could this activity be stably transmitted to subsequent generations in the absence of the *mop1* mutation [36]. To determine if the somatic activity we observed after heat stress can be transmitted to the next generation, we crossed the heat-treated *mop1* homozygous plants that carried transcriptionally reactivated *MuDR* (designated *MuDR~*) and the sibling *mop1* homozygous *MuDR\** control plants, to a tester that was homozygous wild-type for *mop1* and that lacked *MuDR* (Fig 6A). MURA, the protein encoded by *mudrA* causes excision of a reporter element at the *a1-mum2* allele of the *A1* gene, resulting pale kernels with spots of colored revertant tissue. All plants used in these experiments were homozygous for *a1-mum2*. If *mudrA* were fully heritably reactivated, a cross between a *MuDR~/-; mop1/mop1* plant and a tester would be expected to give rise to 50% spotted kernels, and this phenotype would be expected to cosegregate with the reactivated *MuDR* element (Fig 6A). The progeny of ten independent heat-reactivated individuals gave a total of

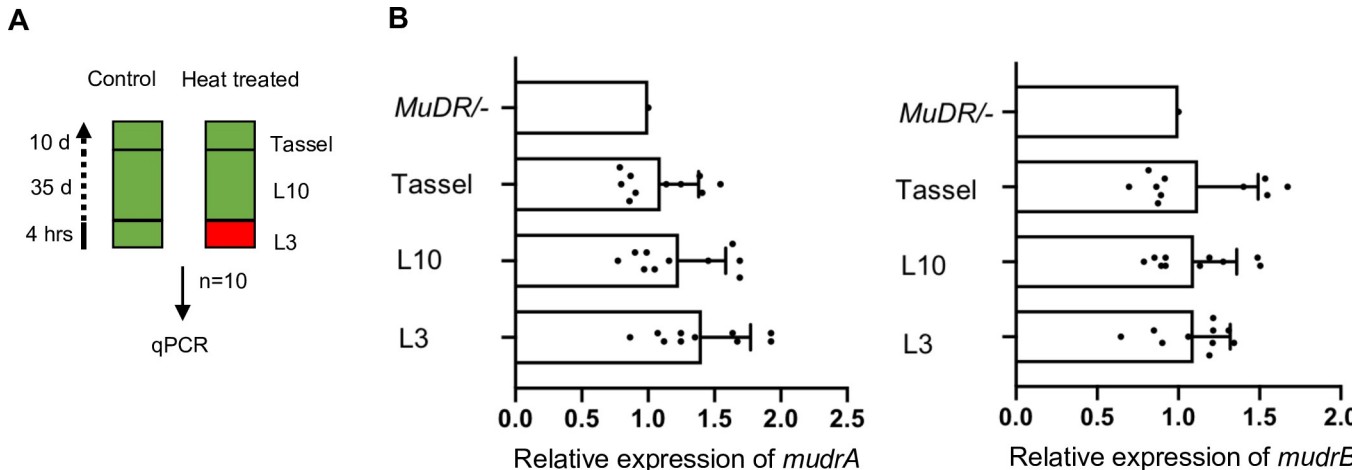

**Fig 5. Expression of *mudrA* and *mudrB* in new emerging tissues following heat stress.** (A) Diagram of the experiment. (B) qPCR was performed to measure transcript levels of *mudrA* and *mudrB* using expression of *Tub2* as an internal control. Expression levels were normalized to that of an active *MuDR* element. Error bars indicate mean ± standard deviation (SD) of the ten biological replicates.

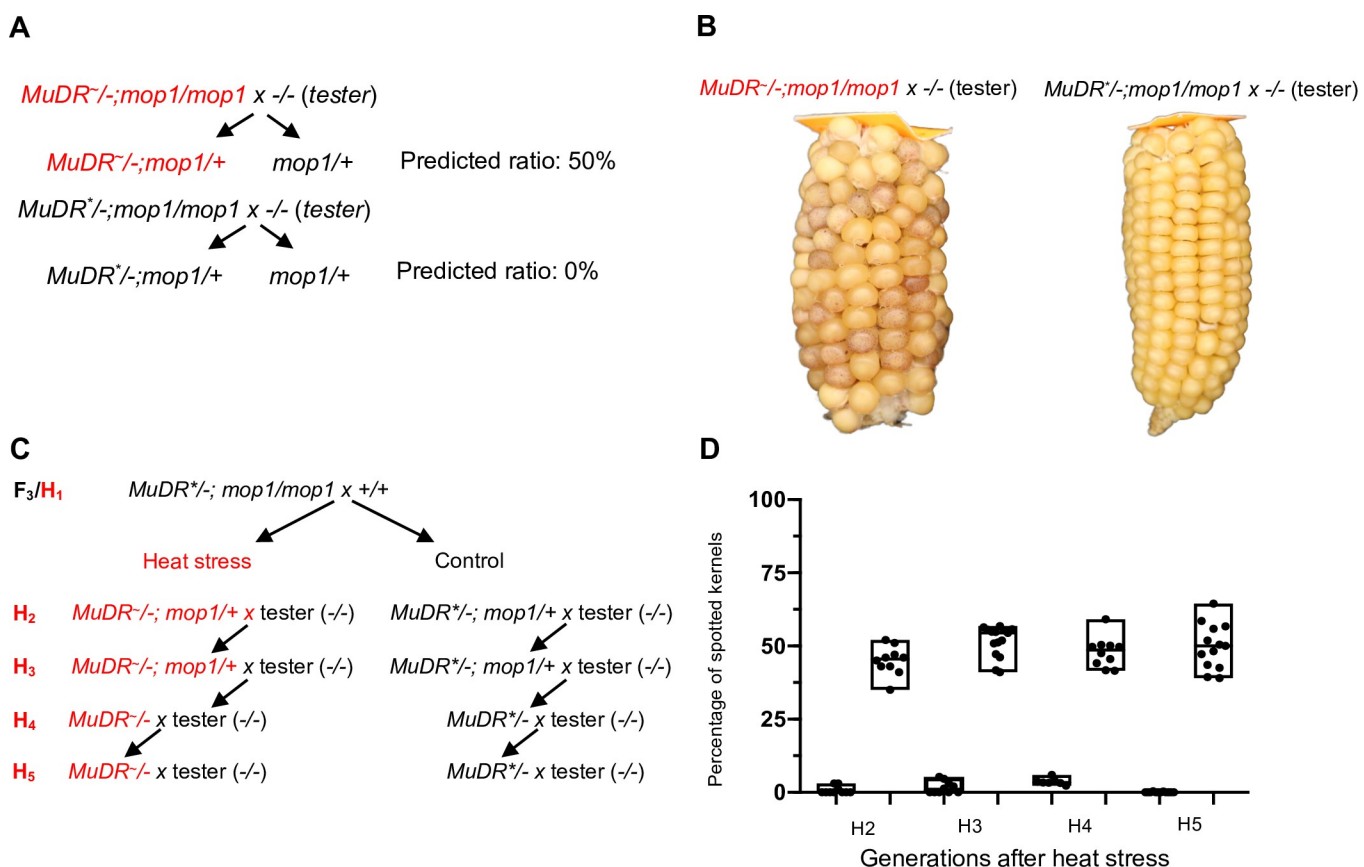

**Fig 6. Testing transgenerational inheritance.** (A) A schematic diagram showing the crosses used to determine transgenerational inheritance. (B) Ears derived from heat-treated and control individuals. (C) Crosses done in the generations following heat stress. (D) Ratios of spotted kernels in subsequent generations following the heat stress ($H_1$) generation. Red text indicates a sample that has been heat-reactivated.

45% spotted kernels. In contrast, ten *mop1* homozygous siblings that carried *MuDR\** and that had not been heat-treated gave rise to an average of only 0.7% spotted kernels after test crossing (Fig 6B and S2 Table). These results show that *MuDR* activity induced by heat treatment was transmitted to the next generation. qRT-PCR in both endosperms and embryos of the spotted and pale progeny kernels and genotyping for the presence or absence of *MuDR* at position 1 on chromosome 9L [85] demonstrated that activity was transmitted to both the embryo and the endosperm, and that this activity cosegregated with the single *MuDR* present in these families (S5 Fig). We employed a similar strategy to test stability of heritability (Fig 6C). We crossed three subsequent generations to testers and counted the spotted kernels. We observed that the progeny of heat-reactivated individuals gave a total of 51%, 48% and 47% spotted kernels in the three subsequent generations. In contrast, subsequent generations of the lineage carrying *MuDR\** that had not been heat-treated gave rise to only a small number of weakly spotted kernels (Fig 6D, S2 Table). These results demonstrate that heat reactivation is stable over multiple generations in a non-mutant genetic background, as is silencing in the absence of heat stress.

## DNA hypomethylation is not associated with transgenerational inheritance of activity

We have shown that DNA methylation is not reduced under heat stress at TIRB, and that even a complete absence of methylation of TIRA under normal conditions does not result in

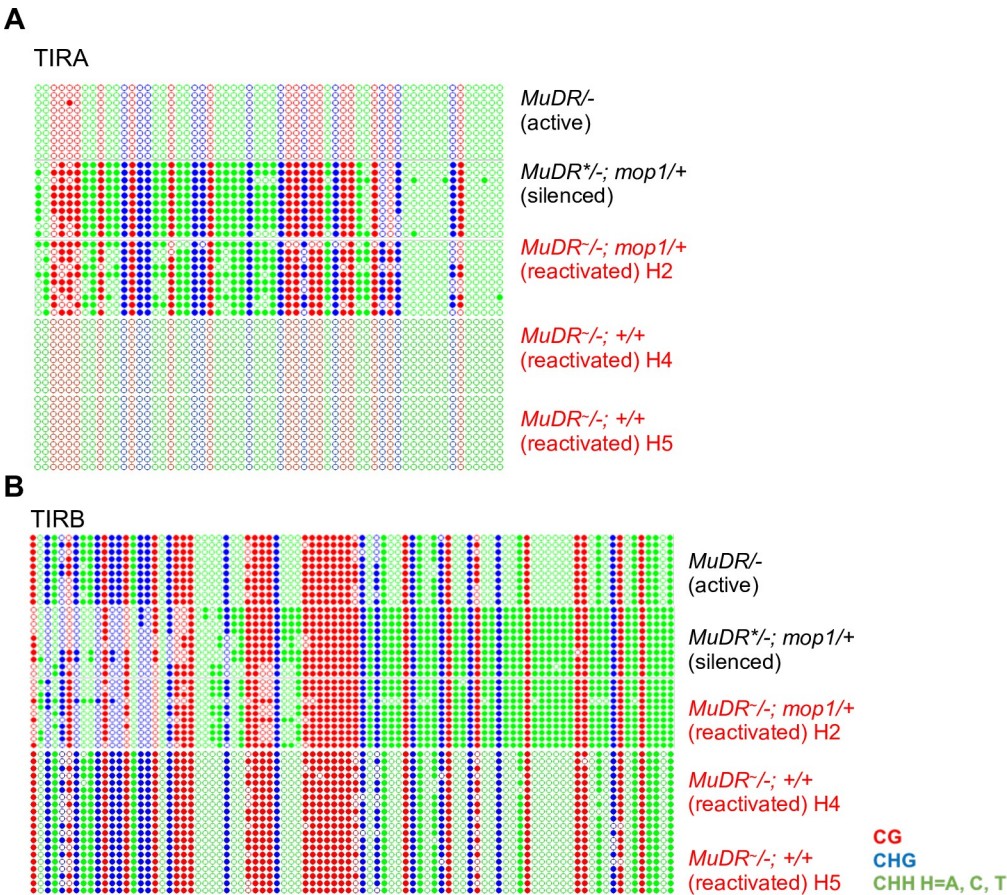

**Fig 7. DNA methylation patterns at TIRA and TIRB in H₂, H₄ and H₅ progeny of heat-treated plants.** (A) DNA methylation patterns at TIRA. (B) DNA methylation patterns at TIRB. Ten individual clones were sequenced from each amplification of bisulfite-treated sample. The cytosines in different sequence contexts are represented by different colors (red, CG; blue, CHG; green, CHH, where H = A, C, or T). Red text indicates plants derived from heat-treated plants. *MuDR** refers to a silenced *MuDR* element. *MuDR~* refers to a reactivated element. For each assay, six independent samples were pooled together.

transcriptional activation. These results suggest that, at least under normal conditions, DNA methylation of *MuDR* is neither necessary nor sufficient to mediate silencing. However, only plants that were *mop1* mutant and whose TIRs were missing either methylation of cytosines in all sequence contexts in the case of TIRA or those in the CHH sequence context in the case of TIRB were reactivated under heat stress. This suggests that a loss of methylation may be a pre-condition for initiation, and perhaps propagation, of continued activity after that stress. To test the latter possibility, we examined DNA methylation at TIRA and TIRB in the *mop1* heterozygous H₂ progenies of heat-reactivated *mop1* mutant plants and those of their unheated *mop1* mutant sibling controls (S1 Fig). Surprisingly, we found that both TIRA and TIRB were extensively methylated in all three sequence contexts in all progenies examined regardless of their activity status (Fig 7). Indeed, their methylation was indistinguishable from that observed at silenced *MuDR* elements. This suggests that although the restoration of MOP1 function does result in the restoration of methylation at both TIRA and TIRB in these heritably reactivated *MuDR* elements, this methylation is not sufficient for reestablishment of silencing at either of these TIRs. In order to determine whether DNA methylation we observed in these wild-type H₂ plants was stable, we examined TIRA and TIRB methylation in plants three and

four generations removed from the initial heat stress. Surprisingly, we found that the observed patterns of methylation in this generation at both TIRs closely resembled that of fully active *MuDR* elements (Fig 7). This suggests that patterns of methylation consistent with activity are in fact restored in the heat stressed lineage after MOP1 function is restored, but only after changes in H3K9me2 at TIRA and in H3K27me3 at TIRB have already taken place.

## Transgenerational heritability of activity is associated with heritability of histone modifications

DNA hypomethylation is not associated with transgenerational inheritance of *MuDR* activity, and DNA hypermethylation does not result in a restoration of silencing in wild-type progeny of heat reactivated mutants. A plausible alternative is that the observed changes in histone marks mediate heritable propagation of activity of both *mudrA* and *mudrB* independent of methylation status. To test this hypothesis, we determined the levels of H3K9me2, H3K27me3 and H3K4me3 at TIRA and TIRB in the *mop1* heterozygous $H_2$ progenies of heat-reactivated *MuDR~/-; mop1/mop1* plants and those of their sibling untreated *MuDR\*/-; mop1/mop1* sibling controls. Consistent with the continued activity of *mudrB* in the progeny of the heat stressed plants, relative levels of H3K27me3 levels remained low and H3K4me3 remained high at TIRB in these plants, suggesting that heritable propagation of H3K27me3 is responsible for that continued activity (Fig 8). Similarly, at TIRA, H3K9me2 remained low and H3K4me3 remained high in these progenies. Interestingly, the increase in DNA methylation in these *MuDR* active *mop1* heterozygous plants was associated with a further decrease in levels of H3K9me2 at TIRA relative to that of their heat stressed *mop1* homozygous parents, down to the levels of the active *MuDR* control. This suggests that an increase in methylation of these active elements in the wild-type background resulted in a concomitant decrease in H3K9me2 at TIRA.

## Discussion

### DNA methylation is neither necessary nor sufficient for the maintenance of silencing at TIRA or TIRB

Our results demonstrating that methylation is not necessary for maintenance of epigenetic silencing in *mop1* mutant plants (Fig 1) and is not sufficient to trigger silencing in $H_2$ reactivated plants (Fig 7) suggest that at this particular locus, DNA methylation is not the key determinative factor with respect to either silencing or its reversal. In contrast, changes in H3K9me2 are closely correlated with changes in TIRA activity, suggesting that it is this modification, rather than DNA methylation, that mediates both activity and heritable transmission of silencing of *mudrA*. Given that H3K9me2 is normally tightly associated with cytosine methylation, particularly in the CHG context [21,93], this result is unexpected. However, our results clearly demonstrate that this modification can be heritably propagated in the absence of DNA methylation and in the absence of the original trigger for silencing, *Muk*. Even more unexpected is our observation that, once *mudrA* becomes silenced, in *mop1* mutants there appears to be reciprocal relationship between DNA methylation of TIRA and H3K9me2 enrichment. Methylation in all three-sequence-context is eliminated throughout TIRA in *mop1* mutants, but this does not result in reactivation of *mudrA*. Instead, H3K9me2 actually significantly *increases* in the *mop1* mutant. This suggests that silencing at this locus is maintained via a balance between DNA and histone methylation, such that a loss of DNA methylation actually triggers an increase in histone modification. This in turn suggests that the state of activity of *mudrA* in some way determines the balance between histone and DNA modification, since

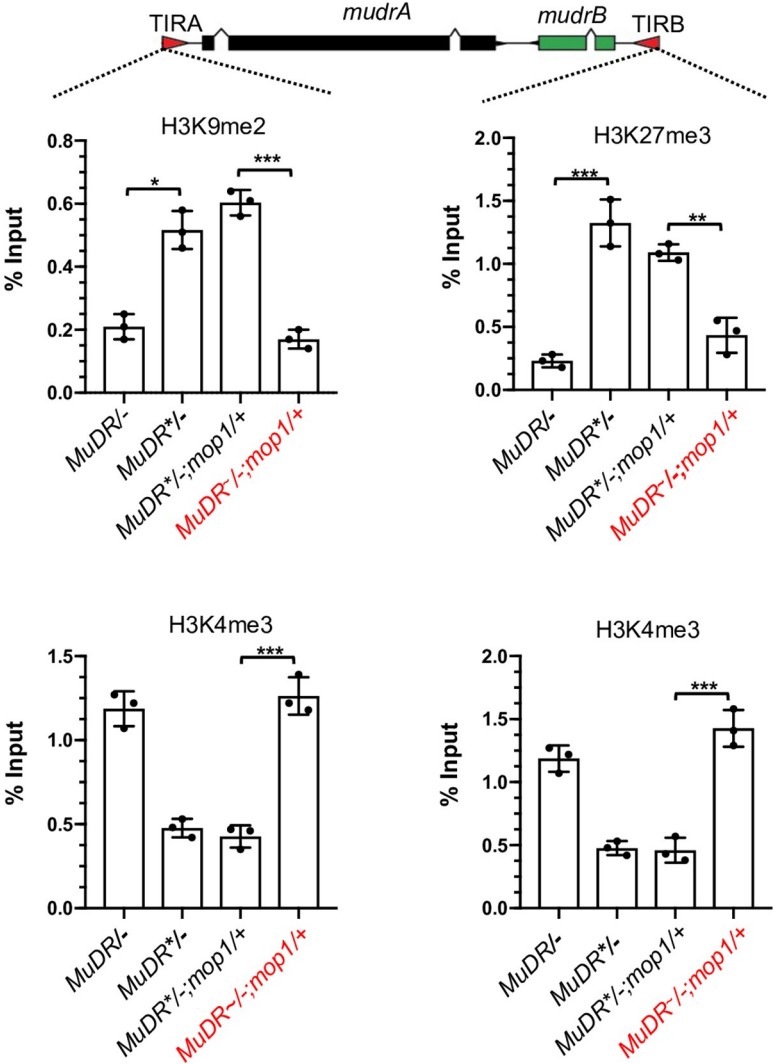

**Fig 8. ChIP-qPCR analysis of enrichment of histone marks, H3K9me2, H3K27me3 and H3K4me3 at TIRA and TIRB.** Relative enrichment of H3K9me2, H3K27me3 and H3K4me3 at TIRA and TIRB in leaf 3 of plants of the indicated genotypes. qPCR signals were normalized to *Copia* and then to the value of input samples. Red text indicates plants derived from heat-treated plants. *MuDR*\* refers to a silenced *MuDR* element. *MuDR˜* refers to a reactivated element. An unpaired t-test was performed. Error bars indicate mean ± standard deviation (SD) of the three biological replicates. \*P<0.05; \*\*P < 0.01; \*\*\*P<0.001.

neither modification by itself appears to be determinative. Our heat experiment supports this hypothesis. Heat rapidly reduces histone modification, but only back down to the level of the silent *mop1* heterozygous siblings rather that to the level of TIRA in an active element. In this case, the combination of an absence of DNA methylation with this reduced level of H3K9me2 appears to be sufficient to permit transcription of *mudrA*, as well as somatic propagation of the reactivated state to daughter cells after the heat is removed. Also supporting a balance hypothesis is the observation that in reactivated *mop1* heterozygous progeny of *mop1* homozygous heat-treated plants, methylation is restored to that observed in silenced elements and levels of H3K9me2 are then reduced to the level observed in active elements. This again suggests that levels of DNA and histone modification balance each other, such that in increase in methylation in the wild-type progeny of reactivated *mop1* mutant plants results in a concomitant

decrease in histone modification. Interestingly, however, at some point after leaf 3 of the H$_2$ generation methylation levels are reduced to those of active *MuDR* elements, suggesting that this reduced methylation level is a consequence, rather than a cause, of maintenance of activity. Collectively, these data suggest that DNA methylation can be a lagging indicator that is responding to a given epigenetic state, rather than determining it.

There are other instances in which silencing can be reversed without a loss of methylation. For instance, mutations in the putative chromatin remodeler *MOTHER OF MORPHEOUS1* (*MOM1*) can result in activation of silenced transgenes and some endogenous loci in the absence of a loss of DNA methylation [94–96]. Similarly, *Microrchidia* (*MORC*) ATPase genes, as well the H3K27 monomethyltransferases, *TRITHORAX-RELATED PROTEIN 5* (*ATXR5*) and *ATXR6* in Arabidopsis, are required for heterochromatin condensation and TE silencing but not for DNA methylation or histone modification associated with that silencing [97–99]. However, unlike reactivated *MuDR* elements in our experiments, reintroduction of the wild-type *MOM1* or *MORC* alleles result in immediate re-silencing. Finally, mutations in two closely related Arabidopsis genes, *MAINTENANCE OF MERISTEMS-LIKE 1* (*MAIL1*) and *MAINTENANCE OF MERISTEMS* (*MAIN*), can also result in activation of a subset of Arabidopsis TEs in the absence of a loss of methylation [100].

## The RdDM pathway buffers the effects of heat stress on silenced *MuDR* elements

Heat stress rapidly reverses silencing and is associated with a reduction of H3K9me2, but only in a *mop1* mutant background. This suggests that although DNA methylation is not required for the maintenance of silencing of *mudrA* and is not sufficient to trigger *de novo* silencing of this gene, it is required to prevent a response to heat stress. Thus, we suggest that the primary role of DNA methylation in this instance is to buffer the effects of heat. We note that this observation is similar but distinct from what has been observed for the *ONSEN* retrotransposon in Arabidopsis. In that case, although heat stress by itself can induce transcription of *ONSEN* [9,75], it is only when the RdDM pathway is deficient that new insertions are transmitted to the next generation. However, in wild-type progenies of heat stressed mutants, *ONSEN* elements are rapidly re-silenced [101]. In contrast, reactivated *MuDR* elements remain active for at least five generations, despite the fact that the RdDM pathway rapidly restores DNA methylation at both TIRA and TIRB. This is likely due to differences between these two elements with respect to the means by which the two elements are maintained in a silenced state. In the absence of *Muk*, *MuDR* elements are stably active over multiple generations [85,102]. This suggests that silencing of *MuDR* requires aberrant transcripts that are distinct from those produced by *MuDR* that are not present in the minimal *Mutator* line. Experiments involving some low copy number elements in Arabidopsis that are activated in the DNA methylation deficient *ddm1* mutant background suggest that the same is true for these elements as well; once activated, these elements remain active even in wild-type progeny plants [103]. In contrast, evidence from other TEs suggests that transcripts from these elements or their derivatives contribute to their own silencing [47,104,105].

## The effects of heat on *mop1* mutants are dependent on the stage of development

Our heat stress experiments demonstrated that although heat exposure has a rapid and dramatic effect on *MuDR* activity in juvenile leaves, heat stress later during adult growth has no effect on this element. Expression analysis of *Hsp90*, a key marker of heat stress in maize, suggests that the older maize *mop1* mutant plants are in fact responding to the heat, but the

response does not include reactivation of *MuDR*. The reason for this difference is not clear. Presumably there are factors expressed later during development that can compensate for the lack of MOP1 in these later leaves. Expression analysis shows dramatic differences between juvenile and adult leaves, including differences in a large number of genes related to stress response [106]. Further, the transition from juvenile to adult growth in maize is associated with a transient loss of *mudrA* silencing in $F_1$ plants carrying both *MuDR* and *Muk*, suggesting that this transition represents an important stage of development with respect to silencing pathways [86]. Future experiments will focus on mutations that affect the time of the juvenile to adult transition that are known to affect the transient loss of *MuDR* silencing.

## Heritably transmitted silencing of TIRB is associated with H3K27me3

Our observation that transgenerationally heritable silencing of *mudrB* is associated with H3K27me3 was surprising, given that this mark is generally associated with somatic silencing of genes that is reset each generation [107]. However, in the absence of that resetting, silencing can be heritably transmitted to the next generation [60,62]. Our data clearly shows that this is the case for *mudrB*, whose H3K27me3 enrichment can be heritably transmitted following the loss of *Mu killer* through at least two rounds of meiosis, and we have evidence that *mudrB* remains stably silenced for at least eight generations [36]. Given that there is no selective pressure to reset TE silencing mediated by H3K27me3, this is not surprising. Interestingly, although most mutants that affect paramutation are, like *Mop1*, components of the RdDM pathway [34], *Maintain repression 12* (*Rmr12*), is a gene encoding a protein orthologous to PICKLE, a putative CHD3-type chromatin remodeling factor in Arabidopsis [108]. In that species, PICKLE can either (directly) reduce or (indirectly) enhance H3K27me3 at target genes. Given that *Rmr12* is required for stable silencing of the paramutable *Pl'* epiallele of the *Pl1* gene, perhaps via modification of H3K27me3, it will be interesting to see whether or not *Rmr12* is required for stable maintenance of silencing of *mudrB*.

There is evidence that heat stress can heritably reverse H3K27me3 at specific loci. H3K27 trimethylation can be reversed by the H3K27me3 demethylase RELATIVE OF EARLY FLOWERING 6 (REF6), which acts in conjunction the chromatin remodeler BRAHMA (BRM) to relax silencing at loci containing CTCTGYTY motifs [109]. In Arabidopsis, under heat stress, HEAT SHOCK TRANSCRIPTION FACTOR A2 (HSFA2) activates *REF6*, which can in turn de-repress *HSFA2* by reducing H3K27me3 at this gene. This feedback loop can extend to the progeny of heat stressed plants, resulting in a heritable reduction in levels of H3K27me3 at REF6-targeting genes [110,111]. However, as in the case for all transgenerational shifts in gene expression, the effect is temporary, and both H3K27me3 and gene expression levels are restored to their original state after two generations.

It should also be noted that at both TIRA and TIRB, H3K4me3 is the most reliable indicator of activity. Thus, it is possible that stable maintenance of activity of both *mudrA* and *mudrB* is mediated via heritable maintenance of H3K4me3. Indeed, an active epiallele derived from callus culture in rice was associated with decrease in DNA methylation and H3K27me3 and an increase in H3K4me3. Overexpression of histone H3K4 *demethylase JUMONJI C 703* (*JMJ703*) resulted in a loss of activity of the reactivated epiallele, a restoration of DNA methylation, an increase in H3K27me3 and a decrease in H3K4me3 [112]. This same H3K4 demethylase has also been shown to be required for stable silencing and DNA methylation of a number of Rice TEs [113]. These data suggest that silencing requires a stable balance between activating and inactivating chromatin marks as well as DNA methylation, changes in any one of which can result in resetting of epigenetic states.

## Conclusions

Overall, our data suggests that even when examining a single TE in a single organism, a wide variety of epigenetic processes can be seen to play a role in both silencing and its reversal. At TIRA, a loss of DNA methylation in *mop1* mutants is associated with what appears to be a compensatory increase in H3K9me2, which is heritably reversed by a brief exposure to heat. Heritable transmission of a reactivated state of *mudrA* is refractive to a restoration of DNA methylation, which instead appears to adjust over time to reflect that activity rather than to block it. In contrast to *mudrA* (and most other TE genes) heritable *mudrB* silencing is associated with H3K37me3 enrichment, which, like H3K9me2 enrichment at TIRA, is readily and heritably reversed by heat treatment. At both TIRA and TIRB, methylation is neither necessary nor sufficient for silencing, but a lack of MOP1 and an associated loss of DNA methylation at both TIRs does appear to be required to precondition both *mudrA* and *mudrB* for responsiveness to heat, consistent with a role for RdDM in buffering the effects of high temperature in maize. Clearly, these results are primarily phenomenological, as the precise mechanism for the reversal of silencing we observe remains a mystery. However, they do suggest that there is a great deal that we do not yet understand about how silenced states can be maintained and how they can be reversed.

## Materials and methods

### Plant materials

Maize seedlings and adult plants were grown in MetroMix under standard long-day greenhouse conditions at 26˚C unless otherwise noted. The minimal *Mutator* line consists of one full-length functional *MuDR* element and one nonautonomous *Mutator* element, *Mu1*. *Mu killer* (*Muk*), a derivative version of the *MuDR* transposon, can heritably trigger epigenetic silencing of that transposon. *Mutator* activity is monitored in seeds via excisions of a *Mu1* element inserted into the *a1-mum2* allele of the *A1* gene, resulting in small sectors of revertant tissue, or spots, in the kernels when activity is present. When *MuDR* activity is absent, the kernels are pale. All plants described in these experiments are homozygous for *a1-mum2*. Although *MuDR* can be present in multiple copies, all of the experiments described here have a single copy of *MuDR* at position 1 on chromosome 2L [102].

All of the crosses used to generate the materials examined in this paper are depicted in S1 Fig. Active *MuDR/-; mop1/mop1* plants were crossed to *Muk/-; mop1/+* plants. The resulting progeny plants were genotyped using PCR Mix (Syd Labs) to screen for plants that carried *MuDR*, *Muk* and that were homozygous for *mop1*, which were designated $F_1$ plants. $F_1$ plants were then crossed to *mop1* heterozygotes. Progeny plants lacking *Muk* but carrying silenced *MuDR* elements, designated *MuDR\**, were designated $F_2$ *MuDR\** progeny. $F_2$ *MuDR\** progeny that were homozygous for *mop1* were crossed to *mop1* heterozygotes. The resulting $F_3$ plants were genotyped for the presence of *MuDR*. These plants were either homozygous or heterozygous for *mop1*. These $F_3$ plants were those that were used for the heat stress experiments. $H_1$ refers to the first generation of these $F_3$ plants that were subjected to heat stress, with successive generations designated $H_2$, $H_3$, etc. *MuDR* was genotyped using primers Ex1 and RLTIR2. Because Ex1 is complementary to sequences flanking *MuDR* in these families, this primer combination is specific to the single *MuDR* element segregating in these families. *Muk* was genotyped using primers TIRAout and 12-4R3. The *mop1* mutation was genotyped using primers ZmRDR2F, ZmRDR2R and TIR6. All primer sequences are provided in S1 Table.

## Tissue sampling

Plants used in all experiments were genotyped individually. The visible portion of each developing leaf blade, when it was ≈10 cm, was harvested when it emerged from the leaf whorl. Only leaf blades of mature leaves were harvested. For the heat reactivation experiment, seedlings were grown at 26˚C for 14 days with a 12–12 light dark cycle. Seedlings were incubated at 42˚C for 4 hours and leaf 3 was harvested immediately after stress treatment. As a control, leaf 3 was also collected from sibling seedlings grown at 26˚C. For each genotype and treatment, 12 biological replicates were used, all of which were siblings. Samples were stored in -80˚C. After sample collection, all seedlings were transferred to a greenhouse at 26˚C. In order to determine if reactivation could be propagated to new emerging tissues, leaf 10 at a similar stage of development (~10 cm, as it emerged from the leaf whorl) and the immature tassel (~20 cm) were collected from each individual (Fig 5A). To determine if the application of heat stress at a later stage of plant development can promote reactivation, an independent set of these seedlings from the same family were used. A similar strategy was employed. However, in this case, seedlings were heat stressed for 4 hours after the plants had grown 28 days at 26˚C. Leaf 7 was collected instead (Fig 3A). For the bisulfite sequencing experiment, leaf 3 was collected from each individual, when it was ≈10 cm, as it emerged from the leaf whorl. In order to minimize potential variation among different individuals, leaves from 6 individuals with the same genotype and treatment were pooled together. For the ChIP assays, a total of ~ 2 g of leaves from leaf 3 of 6 sibling plants with the indicated genotypes was harvested. Three independent sets of these sample collections were collected and analyzed for each genotype and treatment. Leaf samples were fixed with 1% methanol-free formaldehyde and then stored in -80˚C.

## RNA isolation and qRT-PCR analysis

Total RNA was extracted using TRIzol reagent (Invitrogen) and purified by Zymo Direct-zol RNA Miniprep Plus kit. 2 μl of total RNA was first loaded on a 1% agarose gel to check for good quality. Then, RNA was quantified by a NanoDrop spectrophotometer (Thermo Fisher Scientific) and reverse transcribed using an oligo-dT primer and GoScript Reverse Transcriptase (Promega). Quantitative RT-PCR was performed by using SYBR Premix Ex Taq (TaKaRa Bio) on an ABI StepOnePlus Real-Time PCR thermocycler (Thermo Fisher Scientific) according to the manufacturer's instructions. Relative expression values for all experiments were calculated using *Tub2* (Zm00001d010275) as an internal control and determined by using the comparative CT method. Sequences for all primers used for qRT-PCR are available in S1 Table and numerical data used for the figures are available in S4 Table.

## Genomic bisulfite sequencing

These experiments were performed as previously described [87]. In brief, genomic DNA was isolated and digested with RNase A (Thermo Fisher Scientific). 2 μl of this DNA was loaded on a 1% agarose gel to check for good quality and then quantified using a Qubit fluorometer (Thermo Fisher Scientific). 0.5–1 μg of genomic DNA from each genotype and treatment were used for bisulfite conversion. The EZ DNA Methylation-Gold kit (Zymo Research) was used to perform this conversion. Fragments from TIRA and TIRB were PCR-amplified using Epi-Mark Hot Start Taq DNA Polymerase (New England BioLabs). For TIRA, the first amplification was for 20 cycles using p1bis2f and TIRAbis2R with an annealing temperature of 48˚C, followed by re-amplification for 17 cycles using TIRAbis2R and TIRAmF6 with an annealing temperature of 50˚C. Amplicons from TIRB were amplified for 30 cycles using methy_TIRBF and methy_TIRBR with an annealing temperature of 55˚C. The resulting fragments were purified and cloned into pGEM-T Easy Vector (Promega). Ligations and transformations were

performed as directed by the manufacturer's instructions. The resulting colonies were screened for the presence of insertions by performing a colony-based PCR using primers of pGEMF and pGEMTR with an annealing temperature of 52˚C. The sequences of all primers are provided in S1 Table. Plasmid was extracted from positive colonies using the Zyppy Plasmid Kit (Zymo Research). Plasmid from at least 10 independent clones were sequenced at Purdue Genomics Core Facility. The sequences were analyzed using Kismeth (http://katahdin.mssm.edu/kismeth/revpage.pl) [114]. Sequences for all primers are available in S1 Table and numerical data used for figures are available in S3 Table.

## Chromatin immunoprecipitation (ChIP)

The ChIP assay was performed as described previously with some modifications [115–117]. Briefly, leaf samples were treated with 1% methanol-free formaldehyde for 15 minutes under vacuum. Glycine was added to a final concentration of 125 mM, and incubation was continued for 5 additional minutes. Plant tissues were then washed with distilled water and homogenized in liquid nitrogen. Nuclei were isolated and resuspended in 1 mL nuclei lysis buffer (50 Mm tris-HCl pH8, 10 mM EDTA, 0.25% SDS, protease inhibitor). 50 μl of nuclei lysis was harvested for a quality check. DNA was sheared by sonication (Bioruptor UCD-200 sonicator) sufficiently to produce 300 to 500 bp fragments. After centrifugation, the supernatants were diluted to a volume of 3 mL in dilution buffer (1.1% Triton X-100, 1.2 mM EDTA, 16.7 mM Tris-HCl pH8, 167 mM NaCl). Each sample of supernatant was sufficient to make 6 immunoprecipitation (IP) reactions. Every 500 μl sample was precleared with 25 μl protein A/G magnetic beads (Thermo Fisher Scientific) for 1 hour at 4˚C. After the beads were removed using a magnet, the supernatant was removed to a new pre-chilled tube. 50 μl from each sample was used to check for sonication efficiency and set aside to serve as the 10% input control. Antibodies used were anti-H3K9me2 (Millipore), H3K27me2 (Millipore), H3K27me3 (Active Motif), H3K4me3 (Millipore) and H3KAc (Millipore). After incubation overnight with rotation at 4˚C, 30 μl of protein A/G magnetic beads was added and incubation continued for 1.5 hours. The beads were then sequentially washed with 0.5 mL of the following: low salt wash buffer (20 mM Tris (pH 8), 150 mM NaCl, 0.1% (wt/vol) SDS, 1% (vol/vol) Triton X-100, 2 mM EDTA), high salt wash buffer (20 mM Tris (pH 8), 500 mM NaCl, 0.1% (wt/vol) SDS, 1% (vol/vol) Triton X-100, 2 mM EDTA), LiCl wash buffer (10 mM Tris (pH 8), 250 mM LiCl, 1% (wt/vol) sodium deoxycholate, 1% (vol/vol) NP-40 substitute, 1 mM EDTA), TE wash buffer (10 mM Tris (pH 8), 1 mM EDTA). After the final wash, the beads were collected using a magnet and resuspended with 200 μl X-ChIP elution buffer (100 mM NaHCO3, 1% (wt/vol) SDS). A total of 20 μl 5M NaCl was then added to each tube including those samples used for quality checks. Cross-links were reversed by incubation at 65˚C for 6 hours. Residual protein was digested by incubating with 20 μg protease K (Thermo Fisher Scientific) at 55˚C for 1 hour, followed by phenol/chloroform/isoamyl alcohol extraction and DNA precipitation. Final precipitated DNA was dissolved in 50 μl TE. Quantitative RT-PCR was performed by using SYBR Premix Ex Taq (TaKaRa Bio) on an ABI StepOnePlus Real-Time PCR thermocycler (Thermo Fisher Scientific) according to the manufacturer's instructions. The primers used in this study are listed in S2 Table. The primers used to detect H3K9 and H3K27 dimethylation of *Copia* retrotransposons and H3K4 trimethylation of actin that were used as internal controls in this study have been validated previously [117]. Primers used for TIRA (TIRAR and TIRAUTRR) and TIRB (Ex1 and RLTIR2) were those used previously to detect changes in chromatin at these TIRs [86]. Expression values were normalized to the input sample that had been collected earlier using the comparative CT method. Sequences for all primers are available in S1 Table and numerical data used for figures are available in S5 Table.

## Supporting information

**S1 Fig. Diagram of the crosses and generations used in this study.** F1 refers to the first generation during which *MuDR* was exposed to *Muk*. $H_1$, which corresponds to $F_3$, is the generation in which a brief heat treatment was applied. *MuDR* indicates an active *MuDR* element. *MuDR** indicates an inactive *MuDR* element. *MuDR~* indicates a reactivated *MuDR* element. Red text indicates a sample that has been heat-reactivated.
(TIF)

**S2 Fig. ChIP-qPCR analysis of enrichment of H3K27me2 at TIRA.** Relative enrichment of H3K27me2 at TIRA in leaf 3 of plants of the4 indicated genotypes. The qPCR values were normalized to *Copia* and then to the value of input samples. An unpaired t-test was performed. Error bars indicate mean ± standard deviation (SD) of the three biological replicates. $^*P<0.05$; $^{***}P<0.001$.
(TIF)

**S3 Fig. Real-time PCR analysis of *Hsp90* expression in the indicated tissues.** Quantitative real-time PCR was performed to measure transcript levels of *Hsp90*. *Tub2* is used as an internal control gene. For each data point, two of the twelve replicates are pooled.
(TIF)

**S4 Fig. DNA methylation patterns at TIRB of heat-treated $H_1$ *mop1mop1* plants.** DNA methylation patterns at TIRA and TIRB. Ten individual clones were sequenced from each amplification of bisulfite-treated samples with the indicated genotypes. The cytosines in different sequence contexts are represented by different colors (red, CG; blue, CHG; green, CHH, where H = A, C, or T). For each sample, six independent samples were pooled together.
(TIF)

**S5 Fig. Analysis of *mudrA* and *mudrB* expression in progenies of $H_1$ heat stressed plants.** (A) Genotyping results of an ear from the $H_2$ generation. (B) qRT-PCR analysis of *mudrA* and *mudrB* expression in embryos and endosperms from kernels derived from three independent ears derived from crosses of $H_1$ heat stressed plants and control. *Tub2* is used as an internal control gene. Red text indicates a sample that has been heat-treated. For each family, three independent biological replicates were used.
(TIF)

**S1 Table. Primers used in this paper.**
(XLSX)

**S2 Table. Number of spotted progeny from test crosses.**
(XLSX)

**S3 Table. Summary of all bisulfite sequencing results.**
(XLSX)

**S4 Table. Summary of all qRT-PCR results.**
(XLSX)

**S5 Table. Summary of all ChIP-qPCR results.**
(XLSX)

## Acknowledgments

We thank R. Keith Slotkin for critical reading of the manuscript and Anthony Cannon for performing genetic analysis to test the stability of transgenerational heritability.

## Author Contributions

**Conceptualization:** Wei Guo, Dafang Wang, Damon Lisch.

**Data curation:** Wei Guo.

**Formal analysis:** Wei Guo.

**Funding acquisition:** Damon Lisch.

**Investigation:** Wei Guo, Damon Lisch.

**Methodology:** Wei Guo, Damon Lisch.

**Project administration:** Damon Lisch.

**Resources:** Wei Guo, Dafang Wang.

**Supervision:** Wei Guo, Damon Lisch.

**Validation:** Wei Guo.

**Visualization:** Wei Guo, Damon Lisch.

**Writing – original draft:** Wei Guo, Damon Lisch.

**Writing – review & editing:** Wei Guo, Damon Lisch.

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
