## [Decision Letter · Decision Letter 0]

27 Jan 2021

Dear Dr Lisch, dear Damon,

Thank you very much for submitting your Research Article entitled 'RNA-directed DNA methylation prevents rapid and heritable reversal of transposon silencing under heat stress in Zea mays.' to PLOS Genetics.

The manuscript was fully evaluated at the editorial level and by independent peer reviewers. The reviewers appreciated the attention to an important problem, but raised some concerns about the current manuscript. Based on the reviews, we will not be able to accept this version of the manuscript, but we would be willing to review a revised version. We cannot, of course, promise publication at that time.

As you will see from the detailed comments of the reviewers, they agree that your data about the cooperation of epigenetic factors at a specific transposon under different genetic and environmental conditions revealed interesting and novel insight into the complexity of transposon silencing. The MuDR element is documented as an excellent and well-defined model. It is to be hoped that the findings will prove valid for other elements, but evidence for this and a broader analysis is certainly future work.

However, the reviewers have suggestions and concern that require revising the current manuscript. They request full data availability, state-of-the-art quantification of gene expression and bisulfite conversion control, and access to internal controls for some of the assays. They recommend providing more and better references to previous work on the maize RdDM mutants and the role of the polycomb complex beyond FLC. An important point raised repeatedly is the striking difference of heat stress activation between younger and older developmental stages, which should be explained or at least discussed. The different kinetics of transgenerational methylation adaptation could be strengthened as suggested by data for the intermediate generation.

Text and Figures need several modifications, additions, and careful editing to ensure congruency and correctness.

If you decide to revise the manuscript for further consideration at PLOS Genetics, please aim to resubmit within the next 60 days, unless it will take extra time to address the concerns of the reviewers, in which case we would appreciate an expected resubmission date by email to plosgenetics@plos.org.

[LINK]

We are sorry that we cannot be more positive about your manuscript at this stage. Please do not hesitate to contact us if you have any concerns or questions.

Best regards, Ortrun

Ortrun Mittelsten Scheid

Associate Editor

PLOS Genetics

Wendy Bickmore

Section Editor: Epigenetics

PLOS Genetics

As you will see from the detailed comments of the reviewers, they agree that your data about the cooperation of epigenetic factors at a specific transposon under different genetic and environmental conditions revealed interesting and novel insight into the complexity of transposon silencing. The MuDR element is documented as an excellent and well-defined model. It is to be hoped that the findings will prove valid for other elements, but evidence for this and a broader analysis is certainly future work.

However, the reviewers have suggestions and concern that require revising the current manuscript thoroughly. They request full data availability, state-of-the-art quantification of gene expression and bisulfite conversion control, and access to internal controls for some of the assays. They recommend providing more and better references to previous work on the maize RdDM mutants and the role of the polycomb complex beyond FLC. An important point raised repeatedly is the striking difference of heat stress activation between younger and older developmental stages, which should be explained or at least discussed. The different kinetics of transgenerational methylation adaptation could be strengthened as suggested by data for the intermediate generation.

Text and Figures need several modifications, additions, and careful editing to ensure congruency and correctness.

Reviewer's Responses to Questions

**Comments to the Authors:**

Reviewer #1: The manuscript by Guo et al., “RNA-directed DNA methylation prevents rapid and heritable reversal of transposon silencing under heat stress in Zea mays” shows how MuDR transposon silencing is maintained by overlapping epigenetic mechanisms: DNA methylation, H3K9 and K27 dimethylation (me2) as well as H3K27me3. Several findings presented the manuscript are novel and significant for the field of epigenetics and transposon regulation:

- A single transposable element (TE) displays different epigenetic regulation on its 5’ and 3’ ends (TIRA and TIRB respectively) through histone modifications, H3K9me2 and H3K27me2 on TIRA, and H3K27me3 on TIRB. Although those marks are well associated with transcriptional gene silencing, they have been shown to play different roles (constitutive heterochromatin, developmental silencing,…). Here, the authors illustrate a very clear example of all those marks cooperating to maintain a functional TE under control.

- The authors also show how loss of RdDM and DNA methylation, present at both TIRs together with histone modifications, does not result in the immediate transcriptional reactivation and transposition of MuDR. However, in the absence of DNA methylation MuDR reactivates upon heat stress. Many studies in Arabidopsis have shown that loss of RdDM is largely inconsequential to plant development or genome instability caused by TE reactivation under laboratory standard growth conditions. Here, it is shown how DNA methylation is not only required for the transgenerational stability of the TE silencing, but to prevent the mobilization of MuDR under environmental stresses that might represent a more natural situation.

The manuscript is well written, methods are well described, and the claims are properly contextualized in light of current literature. There are only minor corrections that would improve the quality of the manuscript that are listed below:

- As stated in the data availability policy: "numerical data that underlies graphs or summary statistics should be provided in spreadsheet form as supporting information". I couldn't find any spreadsheet with data relative to qPCR and phenotyping graphs while the authors claim that "all data ara fully available without restrictions". Authors should check with editors whether the raw data should be provided for those figures to be on the safe side.

- The introduction does a very good job at explaining the reader the biological model and working system. However, I have the impression it fails to introduce the biological question that the authors want to address. This is found within the first three sentences of the results section (lines 202-204).

- In lines 208-210, the authors state that “in control plants … all cytosines in TIRA were unmethylated … indicated that bisulfite conversion was efficient”. I disagree, as bisulfite conversion is performed independently for each DNA sample, the only way to address conversion efficiency is to investigate DNA methylation levels from sequences with known methylation (no DNA methylation) within the same DNA sample or spiked unmethylathed DNA before conversion (such as plasmid or lambda DNA). Authors might want to rephrase or eliminate the statement.

- In several instances (Fig.1A, 3B, 3C, S5) expression of MuDR is investigated by RT-PCR. qRT-PCR might have been more appropriated as means to properly investigate subtle effects on MuDR expression.

- Given that the authors have used internal controls for the ChIP-qPCR it would be advisable to add those in the supplementary data as means for the reader to see the intrinsic variability of their ChIP experiments.

Reviewer #2: RNA-directed DNA methylation prevents rapid and heritable reversal of transposon silencing under heat stress in Zea mays.

Guo et al. provides a highly detailed study of the epigenetic regulation related to the activity of a single Mu element in wild type and mop1 mutant maize plants. The work describes the relations of DNA methylation and histone modifications of the element's terminal inverted repeats to transcriptional silencing and reactivation by the Muk element and heat, respectively. The latter relations are determined further in the context of transgenerational inheritance.

A strength of the work is the clearly defined genetic test system. By combining it with RT-PCR, clonal bisulfite sequencing, and ChIP analyses, the authors convincingly demonstrate that loss of DNA methylation in mop1 mutants does not prevent heritable silencing. Instead, the stable silencing in mop1 mutants is associated with the increase of different histone modifications at the two terminal inverted repeats of the Mu element, which mediate promotor function for the respective genes. For TIRB H3K27me3 is newly attributed to transposon silencing in maize.

As the main result the authors demonstrate that in contrast to wild type, missing DNA methylation in mop1 mutants allows for reactivation of the TE by heat treatment, which is associated with a reduction of the respective repressive histone modifications. This result indicates DNA methylation in functioning as a buffer against the effects of heat stress.

The reactivation was seen in leaves of young, 14 day-old, but not in leaves of older, 28 day-old, plants. Given the relatively short time interval between the sampling and the similarity of the tissues, this result is somehow unexpected. Do the older plants respond to the heat treatment as tested by hsp90 expression in the younger ones?

In the analyses of transgenerational stability of the reactivated state for TIRA both MuDR/-; +/+ active and H5 reactivated samples show extreme, fully demethylated DNA. In contrast the heat stressed (H2 reactivated) samples are strongly methylated in all contexts as the silenced MuDR*/-;mop1/+ are. For TIRB similar results were obtained but only related to asymmetric DNA methylation. These results are discussed as an adaption of the methylation state to the activity state of the locus, only after multiple generations. To strengthen this point it would be required to analyze DNA methylation in the generations in between i.e. H3 and H4.

Although this work is a mainly descriptive work on a specific example of a single transposon type exemplar, the indications of this work that DNA methylation can rather respond to than determine the activity of a locus and that the balance of DNA methylation and histone modifications is dependent on transcriptional activity, adds important information on the variety of mechanisms involved in transgenerational silencing of transposable elements in maize. Especially, it points to new unexplored mechanistic relations and independence respectively between DNA methylation and histone modifications in this context. The main statement that RdDM is responsible for the prevention of the transposon silencing reversal is solely built on the involvement of MOP1 in RdDM. The analysis of non-coding RNA could have contributed to a mechanistic understanding of the findings.

Minor comments:

How do the author explain the loss of methylation at the 5' end of the TIRB in MuDR*/-; mop1/+ silenced plants in the generation following Muk exposure?

Line 35: dimethylation instead of demethylation

That Aat is a housekeeping gene that was used as a positive expression control should be mentioned already in the legend of figure 1.

In figure 3A, right scheme L7 instead of L3 was collected according to the main text.

Sentence in line 322 needs rewording

That MuDR~ indicates a reactivated MuDR element should be mentioned already in the legend of figure 4.

In figure 7 B sequence orientation of TIRB is reversed compared to figure 1.

Line 810: Ear ears (doubling)

In the legend of figure 6 description of C is missing

Reviewer #3: In this work, Guo et al. use the maize Mutator system for studying the epigenetic silencing mediated by the RdDM pathway. For doing this, they produced maize plant families segregating for a single silenced MuDR element and homozygous or heterozygous for mop1 (Modifier of paramutation 1 a homolog of RNA DEPENDENT RNA POLYMERASE2, RDR2). In brief, by analyzing MuDR expression, DNA methylation and specific histone modifications (H3K9 and H3K27di-tri methylation) at TIRs of the Mu elements, they observed that loss of methylation does not result in a restoration of that transposon activity; instead, histone modifications are responsible for heritable maintenance of silencing. In addition, in mop1 mutants a short exposure to high temperature rapidly reverses both transcriptional silencing and histone modifications in a heritable way. Based on these observations, they concluded that DNA methylation is not necessary to maintain silencing although, they suggest, it is required to buffer the effects of

the environmental stress on transposable elements.

In general, I find the manuscript not accurately and not clearly written; it contains many typing errors that complicate the reading. There are also some incongruences in the Figures and in particular, some explanations regarding the gels are incomplete.

I also have other concerns on the manuscript content.

When describing the mop1 mutant (lines 91-99) the authors use the definition “largely phenotypically normal”.

This definition is quite ambiguous, especially because the mutant plants are used in heat stress experiments and developmental/genetic defects might indirectly interfere with stress response.

Moreover, much research work has been done and published on maize RdDM pathway and other maize epiregulator mutants, which can be either cited or at least discussed here. They also could be tested for validating the results of this work in different epigenetic background. In particular, the relationship between H3K9 methylation and DNA methylation is not sufficiently described, although this is fundamental to discuss the results of the work presented in the manuscript.

Similarly, I find that citing the peculiar example of AtFLC locus, which is regulated by vernalization and many other epigenetic marks at its chromatin, for describing the role of H3K27me3 modification in gene transcriptional regulation does not give an appropriate and exhaustive information on what is known on this modification on different genomic context.

Regarding the results of the work, my principal and general concern is that the authors investigated the variation in methylation and a few histone modification patterns specifically and only at the MuDr sequences; this approach is quite limiting in studying the regulation of transposons expression by pathways that are known to have a genome-wide role. I think that “control sequences” must be used in the experiments to validate the results obtained for MuDr. Although many genotypes and progenies were characterized for transcriptional activation, DNA methylation and histone modifications, no information on the effect of the MuDr integration on the mopi1 genetic background or of the stress treatments at are genomic loci are reported in the manuscript.

The observation that the combination of heat stress and mop1 mutant reactivated the transposon transcription is quite interesting and very similar to the cited example of Onsen in Arabidopsis. However, the authors observed that the reactivation of MuDR occurred only in young maize plant leaves, while older plants behave differently. No explanations or discussion on these findings were provided by the authors.

After reading the results of this work, we can be convinced that a couple of histone modifications have a principal role in controlling MuDr activation than DNA methylation in maize; however, we miss any evidence and sustainable hypothesis on the epigenomic context and on mechanisms which mediate this activation.

Minor points:

Grammatical errors in lines:

41, 47, 48, 76, 95, 101, 109, 137, 173, 188, 233, 264, 321, 380, 395, 397, 409, 411, 427, 510, 517

In all the gel images please explain the lane and use a ladder. In general, the quality of the figures must be improved.

Fig 3.A MuDR*/- (missing; between mop1/mop1)

Fig 4 is a repetition of Fig3

Reviewer #4: Review of Guo et al., 2020 - RNA-directed DNA methylation prevents rapid and heritable reversal of transposon silencing under heat stress in Zea mays

REVIEWER SUMMARY:

In this manuscript, the authors investigate the relationship between the loss of CHH methylation due to mutation in Mop1 and transcriptional activation of the stably silenced Mutator DNA transposon, MuDR, in maize. They found that loss of mop1 function and CHH methylation at MuDR TIRs does not alleviate silencing of MuDR but results in an increase in H3K9me2 at the 5’ TIR (TIRA) and an increase in H3K27me3 at the 3’ TIR (TIRB) which suggests that these marks are required to maintain silencing. In addition to differences in histone modifications between TIRs, the mop1-dependent DNA methylation pattern differs between TIRs where TIRA loses methylation in all sequence contexts while TIRB only loses CHH methylation suggesting that MuDR is regulated by different epigenetic pathways despite high sequence similarity between TIRs. Stably silenced MuDR (MuDR*) plants homozygous for a mop1 loss of function mutation display reactivation of MuDR upon exposure to a 4-hour heat treatment at the 14-day seedling stage that is associated with reduction of H3K9me2 at TIRA and H3K27me3 at TIRB, with a concomitant increase in H3K4me3. This reactivation is constrained to a specific developmental window and dependent on loss of mop1 function because reactivation is not observed in mop1/+ heterozygotes, +/+ wildtype, or at the 28-day seedling stage in the mop1/- background. The authors show that the heat-dependent activation of MuDR is somatically heritable in the absence of the inducing signal by analyzing leaf 10 and tissue from the tassel which are derived from primordial cells generated well after the stress was applied. This reactivation is also inherited transgenerationally for at least 5 generations in the presence of functional mop1 (mop1/+) and MuDR TIRs maintain lower levels of H3K9me2 and H3K27me3 and high levels of H3K4me3. The authors show that methylation patterns progressively resemble stably active MuDR elements over 4 generations in the presence of functional Mop1, with TIRA progressively losing all methylation and TIRB gaining methylation in all sequence contexts.

SIGNIFICANCE OF FINDINGS:

These results are significant because they dissect a well-characterized epigenetically regulated locus to show that mop1 (and perhaps more generally CHH methylation and RdDM) functions to buffer MuDR silencing from alterations in chromatin modifications (H3K9me2 and H3K27me3) during heat stress. Loss of DNA methylation has been shown to alleviate silencing at many different TEs but the authors clearly show here that MuDR relies on two epigenetic silencing pathways that may have different requirements for DNA methylation. Although this work is focused on a single locus, it is novel because it describes the relationship between loss of methylation and transgenerational inheritance of heat-stress altered chromatin states and transcriptional activation of MuDR. The fact that RdDM is required for preventing heat-induced transgenerational epigenetic changes is fascinating and will require further investigation to determine the generality of these results. This work is important because it increases our understanding of the role of RdDM and builds on previous work investigating stress-responsive transposon activation and the function of Mop1 in maintaining silencing during stress.

OVERALL QUALITY AND PRESENTATION OF THE WORK:

The research methods described are certainly of sufficient quality to draw the authors’ conclusions. However, the presentation of the work, especially the figures, require correction and appear to be incomplete (see reviewer comments). Although the rationale and approach are clear and justified, the manuscript should be carefully edited for sentence structure and brevity before publication.

SPECIFIC COMMENTS:

Abstract/Introduction The authors refer to Mop1 as Modifier of paramutation but the commonly used name is Mediator of paramutation.

Figure 2 Lines under the genotypes indicating silenced versus active are not positioned correctly. (a problem in several figures)

Figure 4 Add to legend that red text is heat stressed and black text is control.

Figure 6 Figure caption is not correct. No caption for 6D, 6C caption incorrect.

Figure 8 Lines under genotypes indicating silences versus active are not positioned correctly.

Figure S3 It is important to show HSP90 expression in tissue-matched non-stressed control plants.

74-78 Pol IV and V are DNA-dependent RNA-polymerases, not DNA polymerases.

Introduction/Discussion The authors need to include discussion of mop1/RdDM in stress response. Although mop1 does not dramatically alter transcription under normal conditions, mop1 has strong effects on ABA-responsive transcriptional programs (Vendramin et al 2020) and loss of rmr6 (Pol IV) also shows defective response to drought (Forestan et al 2016/2019, Lunardon et al., 2016). The authors should also discuss Mikula, Genetics, 1995 which showed that heat-stress alters epigenetic inheritance of R paramutation in maize.

General Reviewer Comments It would be interesting to test the heat-dependent MuDR* reactivation in MuDR*/-;MuKiller/-;mop1/- plants. This would be in the presence of the inducing factor and could provide information about how heat and loss of RdDM affects the establishment of silencing. This is not critical to the relevance of this manuscript.

386-390 The author states that the increase of methylation at TIRA in the active MuDR/-; Mop1 WT lines is associated with the decrease in H3K9me2 but Figure 7 shows that TIRA is unmethylated in active wildtype MuDR lines and is not consistent with the previous paragraph.

Discussion The authors do not discuss the heritable increase in H3K4me3 and how this mark, and the active transcription of the genes associated with it, are relevant to maintaining that activation even in the presence of RdDM and differences in methylation patterns of TIRA and TIRB.

**Have all data underlying the figures and results presented in the manuscript been provided?**

Reviewer #1: **No: **As stated in the data availability policy: "numerical data that underlies graphs or summary statistics should be provided in spreadsheet form as supporting information". I couldn't find any spreadsheet with data relative to qPCR and phenotyping graphs while the authors claim that "all data ara fully available without restrictions". Authors should check with editors whether the raw data should be provided for those figures to be on the safe side.

Reviewer #2: Yes

Reviewer #3: None

Reviewer #4: Yes

PLOS authors have the option to publish the peer review history of their article (what does this mean?). If published, this will include your full peer review and any attached files.

Reviewer #1: No

Reviewer #2: No

Reviewer #3: No

Reviewer #4: No

---

## [Decision Letter · Decision Letter 1]

6 May 2021

Dear Dr Lisch,

Thank you very much for submitting your Research Article entitled 'RNA-directed DNA methylation prevents rapid and heritable reversal of transposon silencing under heat stress in Zea mays.' to PLOS Genetics.

The manuscript was fully evaluated at the editorial level and by the previously involved peer reviewers. As you will see, all reviewers agree that the revised version has considered most of the previous concern and support publication. Beside correction of some language error (reviewer 2 and 4), there is only one point that needs another revision: reviewer 2 found some discrepancy between the new data added and the interpretation in the text. We therefore ask you to modify the manuscript accordingly. 

[LINK]

Yours sincerely,

Ortrun Mittelsten Scheid

Associate Editor

PLOS Genetics

Wendy Bickmore

Section Editor: Epigenetics

PLOS Genetics

Reviewer's Responses to Questions

**Comments to the Authors:**

Reviewer #1: The authors have answered to all my comments and I'm satisfied with the revised version of the manuscript. Therefore, I recommend it for publication.

Reviewer #2: Guo et al. provides a detailed study of the epigenetic regulation related to the activity of a single Mu element in wild type and mop1 mutant maize plants. The indications of this work that DNA methylation can rather respond to than determine the activity of a locus and that the balance of DNA methylation and histone modifications is dependent on transcriptional activity, adds important information on the variety of mechanisms involved in transgenerational silencing of transposable elements in maize. Especially, it points to new unexplored mechanistic relations and independence respectively between DNA methylation and histone modifications in this context.

In the revised version my comments are adequately addressed, except the concerns raised in the context of the transgenerational adaption of the methylation states to the activity states after stress induced reactivation:

In the analyses of transgenerational stability of the reactivated state for TIRA both MuDR/-; +/+ active and H5 reactivated samples show extreme, fully demethylated DNA. In contrast the heat stressed (H2 reactivated) samples are strongly methylated in all contexts as the silenced MuDR*/-;mop1/+ are. For TIRB similar results were obtained but only related to asymmetric DNA methylation. These results are discussed as an adaption of the methylation state to the activity state of the locus, only after multiple generations. To strengthen this point it would be required to analyze DNA methylation in the generations in between i.e. H3 and H4.

The authors now provide additional data for TIRA and TIRB on H4, which show the same full demethylation as in H5. The text is not changed accordingly (Line 377) to mention the analysis of the H4 generation neither the conclusion that multiple rounds of meiosis are likely required to adapt the methylation profile to the active state. Clearly this conclusion is weakened by the new results, but can be resolved only by analysis of H3 and the heat-stressed, reactivated H1, which the authors do not provide. The new results however indicate that a gradual loss of methylation in adaption to the activity state of the elements over multiple generations is less likely and it is questionable whether meiosis is required or related to the phenomenon at all. Alternatively, the adaption of the methylation state might be independent of meiosis and started already in H1. Indeed, although not quantitatively analyzed, the methylation profiles of reactivated H2 show demethylation compared to the silenced H1 generation. The new results should be described in the text and the interpretations adjusted.

Examples of remaining grammatical or typing errors:

Line 386 "latter" instead of "later"

Line 332 "could" is doubled

Line 485 "in maize" instead of "is maize"

Reviewer #3: The authors have taken into consideration all concerns raised by my first report at theoretical level. I think that both introduction and discussion were greatly improved.

I understand that the study of the their system/model in a different genomic background is not that simple as well as it is not the validation of the results for MuDr at histone modification level. However, I am convinced that these validations are essential to confirm their observations.

Reviewer #4: All the major issues are addressed, minor issues are listed below

Line numbers Comment

47 Change out-replicated to out-replicate

145-147 Sentence is confusing, please clarify.

543 Change understanding to understand

566 Remove “for” or “in”

595 Spelling error for collected

**Have all data underlying the figures and results presented in the manuscript been provided?**

Reviewer #1: Yes

Reviewer #2: None

Reviewer #3: Yes

Reviewer #4: Yes

PLOS authors have the option to publish the peer review history of their article (what does this mean?). If published, this will include your full peer review and any attached files.

Reviewer #1: No

Reviewer #2: No

Reviewer #3: No

Reviewer #4: No

---

## [Editor Report · Decision Letter 2]

28 May 2021

Dear Dr Lisch,

We are pleased to inform you that your manuscript entitled "RNA-directed DNA methylation prevents rapid and heritable reversal of transposon silencing under heat stress in Zea mays." has been editorially accepted for publication in PLOS Genetics. Congratulations!

Yours sincerely,

Ortrun Mittelsten Scheid

Associate Editor

PLOS Genetics

Wendy Bickmore

Section Editor: Epigenetics

PLOS Genetics

Comments from the reviewers (if applicable):

**Data Deposition**

http://datadryad.org/submit?journalID=pgenetics&manu=PGENETICS-D-20-01828R2

**Press Queries**

---

## [Editor Report · Acceptance letter]

10 Jun 2021

PGENETICS-D-20-01828R2 

RNA-directed DNA methylation prevents rapid and heritable reversal of transposon silencing under heat stress in Zea mays. 

Dear Dr Lisch, 

We are pleased to inform you that your manuscript entitled "RNA-directed DNA methylation prevents rapid and heritable reversal of transposon silencing under heat stress in Zea mays." has been formally accepted for publication in PLOS Genetics! Your manuscript is now with our production department and you will be notified of the publication date in due course.

With kind regards,

Zsofi Zombor

PLOS Genetics

On behalf of:
